# Fast Epigraphical Projection-based Incremental Algorithms for Wasserstein Distributionally Robust Support Vector Machine

**Jiajin Li**
Department of Systems Engineering and Engineering Management
The Chinese University of Hong Kong
`jjli@se.cuhk.edu.hk`

**Caihua Chen**[*]
School of Management and Engineering
Nanjing University
`chchen@nju.edu.cn`

**Anthony Man-Cho So**
Department of Systems Engineering and Engineering Management
The Chinese University of Hong Kong
`manchoso@se.cuhk.edu.hk`

## Abstract

Wasserstein **D**istributionally **R**obust **O**ptimization (DRO) is concerned with finding decisions that perform well on data that are drawn from the worst-case probability distribution within a Wasserstein ball centered at a certain nominal distribution. In recent years, it has been shown that various DRO formulations of learning models admit tractable convex reformulations. However, most existing works propose to solve these convex reformulations by general-purpose solvers, which are not well-suited for tackling large-scale problems. In this paper, we focus on a family of Wasserstein distributionally robust support vector machine (DRSVM) problems and propose two novel epigraphical projection-based incremental algorithms to solve them. The updates in each iteration of these algorithms can be computed in a highly efficient manner. Moreover, we show that the DRSVM problems considered in this paper satisfy a Hölderian growth condition with explicitly determined growth exponents. Consequently, we are able to establish the convergence rates of the proposed incremental algorithms. Our numerical results indicate that the proposed methods are orders of magnitude faster than the state-of-the-art, and the performance gap grows considerably as the problem size increases.

## 1 Introduction

Wasserstein distance-based distributionally robust optimization (DRO) has recently received significant attention in the machine learning community. This can be attributed to its ability to improve generalization performance by robustifying the learning model against unseen data [13, 22]. The DRO approach offers a principled way to regularize empirical risk minimization problems and provides a transparent probabilistic interpretation of a wide range of existing regularization techniques; see, e.g., [4, 10, 22] and the references therein. Moreover, many representative distributionally robust learning models admit equivalent reformulations as tractable convex programs via strong duality [22, 12, 18, 27, 11]. Currently, a standard approach to solving these reformulations is to use

---

[*]Corresponding author

off-the-shelf solvers such as YALMIP or CPLEX. However, these general-purpose solvers do not scale well with the problem size. Such a state of affairs greatly limits the use of the DRO methodology in machine learning applications and naturally motivates the study of the algorithmic aspects of DRO.

In this paper, we aim to design fast iterative methods for solving a family of Wasserstein distributionally robust support vector machine (DRSVM) problems. The SVM is one of the most frequently used classification methods and has enjoyed notable empirical successes in machine learning and data analysis [25, 23]. However, even for this seemingly simple learning model, there are very few works addressing the development of fast algorithms for its Wasserstein DRO formulation, which takes the form $\inf_w \{\frac{c}{2}\|w\|_2^2 + \sup_{\mathbb{Q} \in B_\epsilon^p(\hat{\mathbb{P}}_n)} \mathbb{E}_{(x,y)\sim\mathbb{Q}}[\ell_w(x,y)]\}$ and can be reformulated as

$$\min_{w,\lambda} \lambda\epsilon + \frac{1}{n}\sum_{i=1}^n \max\left\{1 - w^T z_i, 1 + w^T z_i - \lambda\kappa, 0\right\} + \frac{c}{2}\|w\|_2^2, \text{ s.t. } \|w\|_q \leq \lambda; \quad (1)$$

see [12, Theorem 2] and [22, Theorem 3.11]. Problem (1) arises from the vanilla soft-margin SVM model. Here, $\frac{c}{2}\|w\|_2^2$ is the regularization term with $c \geq 0$; $x \in \mathbb{R}^d$ denotes a feature vector and $y \in \{-1, +1\}$ is the associated binary label; $\ell_w(x,y) = \max\{1 - yw^T x, 0\}$ is the hinge loss w.r.t. the feature-label pair $(x,y)$ and learning parameter $w \in \mathbb{R}^d$; $\{(\hat{x}_i, \hat{y}_i)\}_{i=1}^n$ are $n$ training samples independently and identically drawn from an unknown distribution $\mathbb{P}^*$ on the feature-label space $\mathcal{Z} = \mathbb{R}^d \times \{+1, -1\}$ and $z_i = \hat{x}_i \odot \hat{y}_i$; $\hat{\mathbb{P}}_n = \frac{1}{n}\sum_{i=1}^n \delta_{(\hat{x}_i, \hat{y}_i)}$ is the empirical distribution associated with the training samples; $B_\epsilon^p(\hat{\mathbb{P}}_n) = \{\mathbb{Q} \in \mathcal{P}(\mathcal{Z}) : W_p(\mathbb{Q}, \hat{\mathbb{P}}_n) \leq \epsilon\}$ is the ambiguity set defined on the space of probability distributions $\mathcal{P}(\mathcal{Z})$ centered at the empirical distribution $\hat{\mathbb{P}}_n$ and has radius $\epsilon \geq 0$ w.r.t. the $\ell_p$ norm-induced Wasserstein distance

$$W_p(\mathbb{Q}, \hat{\mathbb{P}}_n) = \inf_{\Pi \in \mathcal{P}(\mathcal{Z}\times\mathcal{Z})} \left\{\int_{\mathcal{Z}\times\mathcal{Z}} d_p(\xi, \xi')\, \Pi(d\xi, d\xi') : \Pi(d\xi, \mathcal{Z}) = \mathbb{Q}(d\xi), \Pi(\mathcal{Z}, d\xi') = \hat{\mathbb{P}}_n(d\xi')\right\},$$

where $\xi = (x,y) \in \mathcal{Z}$, $\frac{1}{p} + \frac{1}{q} = 1$, and $d_p(\xi, \xi') = \|x - x'\|_p + \frac{\kappa}{2}|y - y'|$ is the transport cost between two data points $\xi, \xi' \in \mathcal{Z}$ with $\kappa \geq 0$ representing the relative emphasis between feature mismatch and label uncertainty. In particular, the larger the $\kappa$, the more reliable are the labels; see [22, 15] for further details. Intuitively, if the ambiguity set $B_\epsilon^p(\hat{\mathbb{P}}_n)$ contains the ground-truth distribution $\mathbb{P}^*$, then the estimator $w^*$ obtained from an optimal solution to (1) will be less sensitive to unseen feature-label pairs.

In the works [12, 18], the authors proposed cutting surface-based methods to solve the $\ell_p$-DRSVM problem (1). However, in their implementation, they still need to invoke off-the-shelf solvers for certain tasks. Recently, researchers have proposed to use stochastic (sub)gradient descent to tackle a class of Wasserstein DRO problems [5, 24]. Nevertheless, the results in [5, 24] do not apply to the $\ell_p$-DRSVM problem (1), as they require $\kappa = \infty$; i.e., the labels are error-free. Moreover, the transport cost $d_p$ does not satisfy the strong convexity-type condition in [5, Assumption 1] or [24, Assumption A]. On another front, the authors of [15] introduced an ADMM-based first-order algorithmic framework to deal with the Wasserstein distributionally robust logistic regression problem. Though the framework in [15] can be extended to handle the $\ell_p$-DRSVM problem (1), it has two main drawbacks. First, under the framework, the optimal $\lambda^*$ of problem (1) is found by an one-dimensional search, where each update involves fixing $\lambda$ to a given value and solving for the corresponding optimal $w^*(\lambda)$ (which we refer to as the $w$-subproblem). Since the number of $w$-subproblems that arise during the search can be large, the framework is computationally rather demanding. Second, the $w$-subproblem is solved by an ADMM-type algorithm, which involves both primal and dual updates. In order to establish fast (e.g., linear) convergence rate guarantee for the algorithm, one typically requires a regularity condition on the set of primal-dual optimal pairs of the problem at hand. Unfortunately, it is not clear whether the $\ell_p$-DRSVM problem (1) satisfies such a primal-dual regularity condition.

To overcome these drawbacks, we propose two new epigraphical projection-based incremental algorithms for solving the $\ell_p$-DRSVM problem (1), which tackle the variables $(w, \lambda)$ jointly. We focus on the commonly used $\ell_1$, $\ell_2$, and $\ell_\infty$ norm-induced transport costs, which correspond to $q \in \{1, 2, \infty\}$. Our first algorithm is the incremental projected subgradient descent (ISG) method, whose efficiency inherits from that of the projection onto the epigraph $\{(w, \lambda) : \|w\|_q \leq \lambda\}$ of

Table 1: Convergence rates of incremental algorithms for $\ell_p$-DRSVM

| $q$ | $c$ | Hölderian growth | Step size scheme | Convergence rate |
|---|---|---|---|---|
| $q = 1, \infty$ | $c = 0$ | **Sharp** [8, Theorem 3.5] | $\alpha_{k+1} = \rho\alpha_k, \rho \in (0,1)$ | $\mathcal{O}(\rho^k)$ |
| $q = 1, \infty$ | $c > 0$ | **QG** [28, Proposition 6] | $\alpha_k = \frac{\gamma}{nk}, \gamma > 0$ | $\mathcal{O}(\frac{1}{k})$ |
| $q = 2$ | $c = 0$ | **Sharp (BLR)** | $\alpha_{k+1} = \rho\alpha_k, \rho \in (0,1)$ | $\mathcal{O}(\rho^k)$ |
| | | **Not Known** | $\alpha_k = \frac{\gamma}{n\sqrt{k}}, \gamma > 0$ | $\mathcal{O}(\frac{1}{\sqrt{k}})$ |
| $q = 2$ | $c > 0$ | **QG (BLR)** | $\alpha_k = \frac{\gamma}{nk}, \gamma > 0$ | $\mathcal{O}(\frac{1}{k})$ |
| | | **Not Known** | $\alpha_k = \frac{\gamma}{n\sqrt{k}}, \gamma > 0$ | $\mathcal{O}(\frac{1}{\sqrt{k}})$ |

BLR: The result holds under the assumption of bounded linear regularity (BLR) (see Definition 2).

Not Known: Without BLR, it is not known whether the Hölderian growth condition holds.

the $\ell_q$ norm (with $q \in \{1, 2, \infty\}$). The second is the incremental proximal point algorithm (IPPA). Although in general IPPA is less sensitive to the choice of initial step size and can achieve better accuracy than ISG [16], in the context of the $\ell_p$-DRSVM problem (1), each iteration of IPPA requires solving the following subproblem, which we refer to as the *single-sample proximal point update*:

$$\min_{w,\lambda} \max \left\{1 - w^T z_i, 1 + w^T z_i - \lambda\kappa, 0\right\} + \frac{1}{2\alpha}(\|w - \bar{w}\|_2^2 + (\lambda - \bar{\lambda})^2), \quad \text{s.t. } \|w\|_q \le \lambda. \quad (2)$$

Here, $\alpha > 0$ is the step size, $q \in \{1, 2, \infty\}$, and $\bar{w}, \bar{\lambda}$ are given. By carefully exploiting the problem structure, we develop exceptionally efficient solutions to (2). Specifically, we show in Section 3 that the optimal solution to (2) admits an analytic form when $q = 2$ and can be computed by a fast algorithm based on a parametric approach and a modified secant method (cf. [9]) when $q = 1$ or $\infty$.

Next, we investigate the convergence behavior of the proposed ISG and IPPA when applied to problem (1). Our main tool is the following regularity notion:

**Definition 1 (Hölderian growth condition [6])** *A function $f : \mathbb{R}^m \to \mathbb{R}$ is said to satisfy a* Hölderian growth condition *on the domain $\Omega \subseteq \mathbb{R}^m$ if there exist constants $\theta \in [0, 1]$ and $\sigma > 0$ such that*

$$\text{dist}(x, \mathcal{X}) \le \sigma^{-1}(f(x) - f^*)^\theta, \quad \forall x \in \Omega, \quad (3)$$

*where $\mathcal{X}$ denotes the optimal set of $\min_{x \in \Omega} f(x)$ and $f^*$ is the optimal value. The condition (3) is known as* sharpness *when $\theta = 1$ and* quadratic growth (QG) *when $\theta = \frac{1}{2}$; see, e.g., [7].*

We show that for different choices of $q \in \{1, 2, \infty\}$ and $c \ge 0$, the DRSVM problem (1) satisfies either the sharpness condition or QG condition; see Table 1. With the exception of the case $q \in \{1, \infty\}$, where the sharpness (resp. QG) of (1) when $c = 0$ (resp. $c > 0$) essentially follows from [8, Theorem 3.5] (resp. [28, Proposition 6]), the results on the Hölderian growth of problem (1) are new. Consequently, by choosing step sizes that decay at a suitable rate, we establish, for the first time, the fast sublinear (i.e., $\mathcal{O}(\frac{1}{k})$) or linear (i.e., $\mathcal{O}(\rho^k)$) convergence rate of the proposed incremental algorithms when applied to the DRSVM problem (1); see Table 1.

Lastly, we demonstrate the efficiency of our proposed methods through extensive numerical experiments on both synthetic and real data sets. It is worth mentioning that our proposed algorithms can be easily extended to an asynchronous decentralized parallel setting and thus can further meet the requirements of large-scale applications.

## 2 Epigraphical Projection-based Incremental Algorithms

In this section, we present our incremental algorithms for solving the $\ell_p$-DRSVM problem. For simplicity, we focus on the case $c = 0$ in what follows. Our technical development can be extended to handle the general case $c \ge 0$ by noting that the subproblems corresponding to the cases $c = 0$ and $c > 0$ share the same structure.

To begin, observe that the $\ell_p$-DRSVM problem (1) with $c = 0$ can be written compactly as

$$\min_{\|w\|_q \leq \lambda} \frac{1}{n} \sum_{i=1}^{n} f_i(w, \lambda), \tag{4}$$

where $f_i(w, \lambda) = \lambda\epsilon + \max\{1 - w^T z_i, 1 + w^T z_i - \lambda\kappa, 0\}$ is a piecewise affine function. Since problem (4) possesses the vanilla finite-sum structure with a single epigraphical projection constraint, a natural and widely adopted approach to tackling it is to use incremental algorithms. Roughly speaking, such algorithms select one mini-batch of component functions from the objective in (4) at a time based on a certain cyclic order and use the selected functions to update the current iterate. We shall focus on the following two incremental algorithms for solving the DRSVM problem (1). Here, $k$ is the epoch index (i.e., the $k$-th time going through the cyclic order) and $\alpha_k > 0$ is the step size in the $k$-th epoch.

**Incremental Mini-batch Projected Subgradient Algorithm (ISG)**

$$(w_{i+1}^k, \lambda_{i+1}^k) = \text{proj}_{\{\|w\|_q \leq \lambda\}} \left[ (w_i^k, \lambda_i^k) - \alpha_k g_i^k \right], \tag{5}$$

where $g_i^k$ is a subgradient of $\frac{1}{|B_i|} \sum_{j \in B_i} f_j$ at $(w_i^k, \lambda_i^k)$ and $B_i \subseteq \{1, \ldots, n\}$ is the $i$-th mini-batch.

**Incremental Proximal Point Algorithm (IPPA)**

$$(w_{i+1}^k, \lambda_{i+1}^k) = \arg\min_{\|w\|_q \leq \lambda} \left\{ f_i(w, \lambda) + \frac{1}{2\alpha_k} \left( \|w - w_i^k\|_2^2 + (\lambda - \lambda_i^k)^2 \right) \right\}, \tag{6}$$

where $(w_n^k, \lambda_n^k) = (w_0^{k+1}, \lambda_0^{k+1})$.

Now, a natural question is how to solve the subproblems (5) and (6) efficiently. As it turns out, the key lies in an efficient implementation of the $\ell_q$ norm epigraphical projection (with $q \in \{1, 2, \infty\}$). Indeed, such a projection appears explicitly in the ISG update (5) and, as we shall see later, plays a vital role in the design of fast iterative algorithms for the single-sample proximal point update (6). To begin, we note that the $\ell_2$ norm epigraphical projection $\text{proj}_{\{\|w\|_2 \leq \lambda\}}$ has a well-known analytic solution; see [1, Theorem 3.3.6]. Next, the $\ell_1$ norm epigraphical projection $\text{proj}_{\{\|w\|_1 \leq \lambda\}}$ can be found in linear time using the quick-select algorithm; see [26]. Lastly, the $\ell_\infty$ norm epigraphical projection $\text{proj}_{\{\|w\|_\infty \leq \lambda\}}$ can be computed in linear time via the Moreau decomposition

$$\text{proj}_{\{\|w\|_\infty \leq \lambda\}}(x, s) = (x, s) + \text{proj}_{\{\|w\|_1 \leq \lambda\}}(-x, -s).$$

From the above discussion, we see that the ISG update (5) can be computed efficiently. In the next section, we discuss how these epigraphical projections can be used to perform the single-sample proximal point update (6) in an efficient manner.

## 3  Fast Algorithms for Single-Sample Proximal Point Update (6)

**Analytic solution for $q = 2$.** We begin with the case $q = 2$. By combining the terms $\lambda\epsilon$ and $\frac{1}{2\alpha_k}(\lambda - \lambda_i^k)^2$ in (6), we see that the single-sample proximal point update takes the form (cf. (2))

$$\min_{w, \lambda} \underbrace{\max\{1 - w^T z_i, 1 + w^T z_i - \lambda\kappa, 0\}}_{h_i(w, \lambda)} + \frac{1}{2\alpha}(\|w - \bar{w}\|_2^2 + (\lambda - \bar{\lambda})^2), \text{ s.t. } \|w\|_2 \leq \lambda. \tag{7}$$

The main difficulty of (7) lies in the piecewise affine term $h_i$. To handle this term, let $h_{i,1}(w, \lambda) = 1 - w^T z_i$, $h_{i,2}(w, \lambda) = 1 + w^T z_i - \lambda\kappa$, and $h_{i,3}(w, \lambda) = 0$, so that $h_i = \max_{j \in \{1,2,3\}} h_{i,j}$. Observe that if $(w^*, \lambda^*)$ is an optimal solution to (7), then there could only be one, two, or three affine pieces in $h_i$ that are active at $(w^*, \lambda^*)$; i.e., $\Gamma = |\{j : h_i(w^*, \lambda^*) = h_{i,j}(w^*, \lambda^*)\}| \in \{1, 2, 3\}$. This suggests that we can find $(w^*, \lambda^*)$ by exhausting these possibilities. Due to space limitation, we only give an outline of our strategy here. The details can be found in the Appendix.

We start with the case $\Gamma = 1$. For $j = 1, 2, 3$, consider the following problem, which corresponds to the subcase where $h_{i,j}$ is the only active affine piece:

$$\min_{w, \lambda} h_{i,j}(w, \lambda) + \frac{1}{2\alpha}(\|w - \bar{w}\|_2^2 + (\lambda - \bar{\lambda})^2), \text{ s.t. } \|w\|_2 \leq \lambda. \tag{8}$$

Since $h_{i,j}$ is affine in $(w, \lambda)$, it is easy to verify that problem (8) reduces to an $\ell_2$ norm epigraphical projection, which admits an analytic solution, say $(\hat{w}_j, \hat{\lambda}_j)$. If there exists a $j' \in \{1, 2, 3\}$ such that $h_{i,j'}(\hat{w}_{j'}, \hat{\lambda}_{j'}) > h_{i,j}(\hat{w}_{j'}, \hat{\lambda}_{j'})$ for $j \neq j'$, then we know that $(\hat{w}_{j'}, \hat{\lambda}_{j'})$ is optimal for (7) and hence we can terminate the process. Otherwise, we proceed to the case $\Gamma = 2$ and consider, for $1 \leq j < j' \leq 3$, the following problem, which corresponds to the subcase where $h_{i,j}$ and $h_{i,j'}$ are the only two active affine pieces:

$$\min_{w,\lambda} h_{i,j}(w,\lambda) + \frac{1}{2\alpha}(\|w - \bar{w}\|_2^2 + (\lambda - \bar{\lambda})^2), \quad \text{s.t. } h_{i,j}(w,\lambda) = h_{i,j'}(w,\lambda), \; \|w\|_2 \leq \lambda. \quad (9)$$

As shown in the Appendix (Proposition 6.2), the optimal solution to (9) can be found by solving a univariate quartic equation, which can be done efficiently. Now, let $(\hat{w}_{(j,j')}, \hat{\lambda}_{(j,j')})$ be the optimal solution to (9). If there exist $j, j'$ with $1 \leq j < j' \leq 3$ such that $h_{i,j}(\hat{w}_{(j,j')}, \hat{\lambda}_{(j,j')}) = h_{i,j'}(\hat{w}_{(j,j')}, \hat{\lambda}_{(j,j')}) > h_{i,j''}(\hat{w}_{(j,j')}, \hat{\lambda}_{(j,j')})$ with $j'' \in \{1, 2, 3\} \setminus \{j, j'\}$, then $(\hat{w}_{(j,j')}, \hat{\lambda}_{(j,j')})$ is optimal for (7) and we can terminate the process. Otherwise, we proceed to the case $\Gamma = 3$. In this case, we consider the problem

$$\min_{w,\lambda} \frac{1}{2\alpha}(\|w - \bar{w}\|_2^2 + (\lambda - \bar{\lambda})^2), \quad \text{s.t. } h_{i,1}(w,\lambda) = h_{i,2}(w,\lambda) = h_{i,3}(w,\lambda), \; \|w\|_2 \leq \lambda,$$

which reduces to

$$\min_w \frac{1}{2\alpha}\|w - \bar{w}\|_2^2, \quad \text{s.t. } w^T z_i = 1, \; \|w\|_2 \leq \frac{2}{\kappa}. \quad (10)$$

It can be shown that problem (10) admits an analytic solution $\hat{w}$; see the Appendix (Proposition 6.4). Then, the pair $(\hat{w}, \frac{2}{\kappa})$ is an optimal solution to (7).

**Fast iterative algorithm for $q = 1$.** The high-level idea is similar to that for the case $q = 2$; i.e., we systematically go through all valid subcollections of the affine pieces in $h_i$ and test whether they can be active at the optimal solution to the single-sample proximal point update. The main difference here is that the subproblems arising from the subcollections do not necessarily admit analytic solutions. To overcome this difficulty, we propose a modified secant algorithm (cf. [9]) to search for the optimal dual multiplier of the subproblem and use it to recover the optimal solution to the original subproblem via $\ell_1$ norm epigraphical projection. Again, we give an outline of our strategy here and relegate the details to the Appendix.

To begin, we rewrite the single-sample proximal point update (6) for the case $q = 1$ as

$$\min_{w,\lambda,\mu} \; \mu + \frac{1}{2\alpha}\left(\|w - \bar{w}\|_2^2 + (\lambda - \bar{\lambda})^2\right)$$
$$\text{s.t. } h_{i,j}(w,\lambda) \leq \mu, \; j = 1, 2, 3; \; \|w\|_1 \leq \lambda. \quad (11)$$

For reason that would become clear in a moment, we shall not go through the cases $\Gamma = 1, 2, 3$ as before. Instead, consider first the case where $h_{i,3}$ is inactive. If $h_{i,1}$ is also inactive, then we consider the problem $\min_{w,\lambda} h_{i,2}(w,\lambda) + \frac{1}{2\alpha}\left(\|w - \bar{w}\|_2^2 + (\lambda - \bar{\lambda})^2\right)$, which, by the affine nature of $h_{i,2}$, is equivalent to an $\ell_1$ norm epigraphical projection. If $h_{i,1}$ is active, then we consider the problem

$$\min_{w,\lambda} h_{i,1}(w,\lambda) + \frac{1}{2\alpha}(\|w - \bar{w}\|_2^2 + (\lambda - \bar{\lambda})^2), \quad \text{s.t. } h_{i,2}(w,\lambda) \leq h_{i,1}(w,\lambda), \; \|w\|_1 \leq \lambda. \quad (12)$$

Note that $h_{i,2}$ can be active or inactive, and the constraint $h_{i,2}(w,\lambda) \leq h_{i,1}(w,\lambda)$ allows us to treat both possibilities simultaneously. Hence, we do not need to tackle them separately as we did in the case $q = 2$. Observe that problem (12) can be cast into the form

$$\min_{w,\lambda} \frac{1}{2\alpha}(\|w - \bar{w}\|_2^2 + (\lambda - \bar{\lambda})^2), \quad \text{s.t. } w^T z \leq a\lambda + b \; (\leftarrow \sigma \geq 0), \; \|w\|_1 \leq \lambda, \quad (13)$$

where, with an abuse of notation, we use $\bar{w} \in \mathbb{R}^d$, $\bar{\lambda} \in \mathbb{R}$ here again and caution the reader that they are different from those in (12), and $z = z_i$, $a = \frac{\kappa}{2}$, $b = 0$. Before we discuss how to solve the subproblem (13), let us note that it arises in the case where $h_{i,3}$ is active as well. Indeed, if $h_{i,3}$ is active and $h_{i,1}$ is inactive, then we have $z = z_i$, $a = \kappa$, $b = -1$, which corresponds to the constraint $h_{i,2}(w,\lambda) \leq h_{i,3}(w,\lambda)$ and covers the possibilities that $h_{i,2}$ is active and inactive. On the other hand, if $h_{i,3}$ is active and $h_{i,2}$ is inactive, then we have $z = -z_i$, $a = 0$, $b = -1$, which corresponds

to the constraint $h_{i,1}(w, \lambda) \leq h_{i,3}(w, \lambda)$ and covers the possibilities that $h_{i,1}$ is active and inactive. The only remaining case is when $h_{i,1}, h_{i,2}, h_{i,3}$ are all active. In this case, we consider problem (10) with $\|w\|_2 \leq \frac{2}{\kappa}$ replaced by $\|w\|_1 \leq \frac{2}{\kappa}$. As shown in the Appendix, such a problem can be tackled using the technique for solving (13). We go through the above cases sequentially and terminate the process if the solution to the subproblem in any one of the cases satisfies the optimality conditions of (11).

Now, let us return to the issue of solving (13). The main idea is to perform an one-dimensional search on the dual variable $\sigma$ to find the optimal dual multiplier $\sigma^*$. Specifically, consider the problem

$$\min_{\|w\|_1 \leq \lambda} \quad \frac{1}{2\alpha} \left( \|w - \bar{w}\|_2^2 + (\lambda - \bar{\lambda})^2 \right) + \sigma(w^T z - a\kappa - b). \tag{14}$$

Let $(\hat{w}(\sigma), \hat{\lambda}(\sigma))$ be the optimal solution to (14) and define the function $p : \mathbb{R}_+ \to \mathbb{R}$ by $p(\sigma) = \hat{w}(\sigma)^T z - a\kappa - b$. Inspired by [17], we establish the following monotonicity property of $p$, which will be crucial to our development of an extremely efficient algorithm for solving (13) later.

**Proposition 3.1** *If $\sigma$ satisfies (i) $\sigma = 0$ and $p(\sigma) \leq 0$, or (ii) $p(\sigma) = 0$, then $(\hat{w}(\sigma), \hat{\lambda}(\sigma))$ is the optimal solution to* (13). *Moreover, $p$ is continuous and monotonically non-increasing on $\mathbb{R}_+$.*

In view of Proposition 3.1, we first check if $p(0) \leq 0$ via an $\ell_1$ norm epigraphical projection. If not, then we search for the $\sigma^* \geq 0$ that satisfies $p(\sigma^*) = 0$ by the secant method, with some special modifications designed to speed up its convergence [9]. Let us now give a high-level description of our modified secant method. We refer the reader to the Appendix (Algorithm 1) for details.

At the beginning of a generic iteration of the method, we have an interval $[\sigma_l, \sigma_u]$ that contains $\sigma^*$, with $r_l = -p(\sigma_l) < 0$ and $r_u = -p(\sigma_u) > 0$. The initial interval can be found by considering the optimality conditions of (11) (i.e., $\sigma^* \in [0, 1]$). We then take a secant step to get a new point $\sigma$ with $r = -p(\sigma)$ and perform the update on $\sigma_l, \sigma_u$ as follows.

Suppose that $r > 0$. If $\sigma$ lies in the left-half of the interval (i.e., $\sigma < \frac{\sigma_l + \sigma_u}{2}$), then we update $\sigma_u$ to $\sigma$. Otherwise, we take an auxiliary secant step based on $\sigma$ and $\sigma_u$ to get a point $\sigma'$, and we update $\sigma_u$ to $\max\{\sigma', 0.6\sigma_l + 0.4\sigma\}$. Such a choice ensures that the interval length is reduced by a factor of $0.6$ or less. The case where $r < 0$ is similar, except that $\sigma_l$ is updated. If $r = 0$, then by Proposition 3.1 we have found the optimal dual multiplier $\sigma^*$ and hence can terminate.

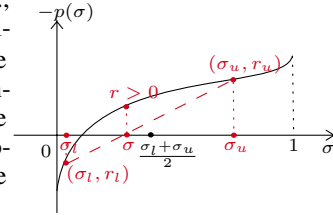

Finally, for the case $q = \infty$, we can follow the same procedure as the case $q = 1$. The details can be found in the Appendix.

## 4 Convergence Rate Analysis of Incremental Algorithms

In this section, we study the convergence behavior of our proposed incremental methods ISG and IPPA. Our starting point is to understand the conditons under which the $\ell_p$-DRSVM problem (1) possesses the Hölderian growth condition (3). Then, by determining the value of the growth exponent $\theta$ and using it to choose step sizes that decay at a suitable rate, we can establish the convergence rates of ISG and IPPA. To begin, let us consider problem (1) with $q \in \{1, \infty\}$. If $c = 0$, then problem (1) satisfies the sharpness (i.e., $\theta = 1$) condition. This follows essentially from [8, Theorem 3.5], as the objective of (1) has polyhedral epigraph and the constraint is polyhedral. On the other hand, if $c > 0$, then since the piecewise affine term in (1) has a polyhedral epigraph and the constraint is polyhedral, we can invoke [28, Proposition 6] and conclude that problem (1) satisfies the QG (i.e., $\theta = \frac{1}{2}$) condition.

Next, let us consider the case $q = 2$. From the above discussion, one may expect that similar conclusions hold for this case. However, as the following example shows, this case is more subtle and requires a more careful treatment.

**Example 4.1** *Consider the problem $\min_{w, \lambda} 0.1\lambda + |1 - w_1|$, s.t. $\sqrt{w_1^2 + w_2^2} \leq \lambda$, which is an instance of* (1) *with $q = 2$, $c = 0$. It is easy to verify that the optimal solution is $w^* = (1, 0)$, $\lambda^* = 1$.*

*Consider feasible points of the form* $(w_1, w_2, \lambda) = (w_1, \sqrt{1 - w_1^2}, 1)$, *which tend to* $(w^*, \lambda^*)$ *as* $w_1$
*tends to 1. A simple calculation yields* $\mathrm{dist}((w_1, w_2, \lambda), (w^*, \lambda^*)) = \sqrt{2|1 - w_1|} = \omega(|1 - w_1|)$,
*which shows that the instance cannot satisfy the sharpness condition.*

As it turns out, it is still possible to establish the sharpness or QG condition for problem (1) with
$q = 2$ under a well-known sufficient condition called *bounded linear regularity*. Let us begin with
the definition.

**Definition 2 (Bounded linear regularity [2, Definition 5.6])** *Let* $C_1, \dots, C_N$ *be closed convex*
*subsets of* $\mathbb{R}^d$ *with a non-empty intersection* $C$. *We say that the collection* $\{C_1, \dots, C_N\}$ *is* bounded
linearly regular *(BLR) if for every bounded subset* $\mathcal{B}$ *of* $\mathbb{R}^d$, *there exists a constant* $\kappa > 0$ *such that*

$$\mathrm{dist}(x, C) \leq \kappa \max_{i \in \{1, \dots, N\}} \mathrm{dist}(x, C_i), \text{ for all } x \in \mathcal{B}.$$

Using the above definition, we can establish the following result; see the Appendix for the proof.

**Proposition 4.2** *Consider problem* (1) *with* $q = 2$. *Let* $\mathcal{X}$ *be the set of optimal solutions and*
$L_2^d = \{(w, \lambda) \in \mathbb{R}^d \times \mathbb{R} : \|w\|_2 \leq \lambda\}$ *be the constraint set. Suppose that* $\mathcal{X} \cap \mathrm{ri}(L_2^d) \neq \emptyset$.
*Consequently, problem* (1) *satisfies the sharpness condition when* $c = 0$ *and the QG condition when*
$c > 0$.

By combining Proposition 4.2 with an appropriate choice of step sizes, we obtain the following
convergence rate guarantees for ISG and IPPA. The proof can be found in the Appendix.

**Theorem 4.3** *Let* $\{x^k = (w_0^k, \lambda_0^k)\}$ *be the sequence of iterates generated by ISG or IPPA.*

*(1) If problem* (1) *satisfies the* sharpness *condition, then by choosing the geometrically diminishing*
*step sizes* $\alpha_k = \alpha_0 \rho^k$ *with* $\alpha_0 \geq \frac{\sigma \, \mathrm{dist}(x^0, \mathcal{X})}{2L^2 n}$ *and* $\sqrt{1 - \frac{\sigma^2}{2L^2}} \leq \rho < 1$, *the sequence* $\{x^k\}$
*converges linearly to an optimal solution to* (1)*; i.e.,* $\mathrm{dist}(x^k, \mathcal{X}) \leq \mathcal{O}(\rho^k)$ *for all* $k \geq 0$.

*(2) If problem* (1) *satisfies the* quadratic growth *condition, then by choosing the polynomially decay-*
*ing step sizes* $\alpha_k = \frac{\gamma}{nk}$ *with* $\gamma > \frac{1}{2\sigma}$, *the sequence* $\{x^k\}$ *converges to an optimal solution to* (1)
*at the rate* $\mathcal{O}(\frac{1}{\sqrt{k}})$ *and* $\{f(x^k) - f^*\}$ *converges to zero at the rate* $\mathcal{O}(\frac{1}{k})$.

*(3) (See [20, Proposition 2.10]) For the general convex problem* (1)*, by choosing the step sizes*
$\alpha_k = \frac{\gamma}{n\sqrt{k}}$ *with* $\gamma > 0$, *the sequence* $\{\min_{0 \leq k \leq K} f(x^k) - f^*\}$ *converges to zero at the rate* $\mathcal{O}(\frac{1}{\sqrt{K}})$.

## 5 Experiment Results

In this section, we present numerical results to demonstrate the efficiency of our proposed incre-
mental methods. All simulations are implemented using MATLAB R2019b on a computer running
Windows 10 with a 3.20 GHz, the Intel(R) Core(TM) i7-8700 processor, and 16 GB of RAM. To
begin, we evaluate our two proposed incremental methods ISG and IPPA in different settings to
corroborate our theoretical results in Section 4 and to better understand their empirical strengths and
weaknesses. Based on this, we develop a hybrid algorithm that combines the advantages of both
ISG and IPPA to further speed up the convergence in practice. Next, we compare the wall-clock
time of our algorithms with GS-ADMM [15] and YALMIP (i.e., IPOPT) solver on real datasets.
For sake of fairness, we only extend the first-order algorithmic framework (referred to as GS-
ADMM) to tackle the $\ell_\infty$-DRSVM problem. In fact, the faster inner solver conjugate gradient
with an active set method can only tackle the $\ell_\infty$ case in [15]. The implementation details to re-
produce all numerical results in this section are given in the Appendix. Our code is available at
https://github.com/gerrili1996/Incremental_DRSVM.

### 5.1 Synthetic data: Different regularity conditions and their step size schemes

Our setup for the synthetic experiments is as follows. First, we generate the learning parameter $w^*$
and feature vectors $\{x_i\}_{i=1}^n$ independently and identically (i.i.d) from the standard normal distribu-
tion $\mathcal{N}(0, I_d)$ and the noisy measurements $\{\xi_i\}_{i=1}^n$ i.i.d from $\mathcal{N}(0, \sigma^2 I_d)$ (e.g., $\sigma = 0.5$). Then, we

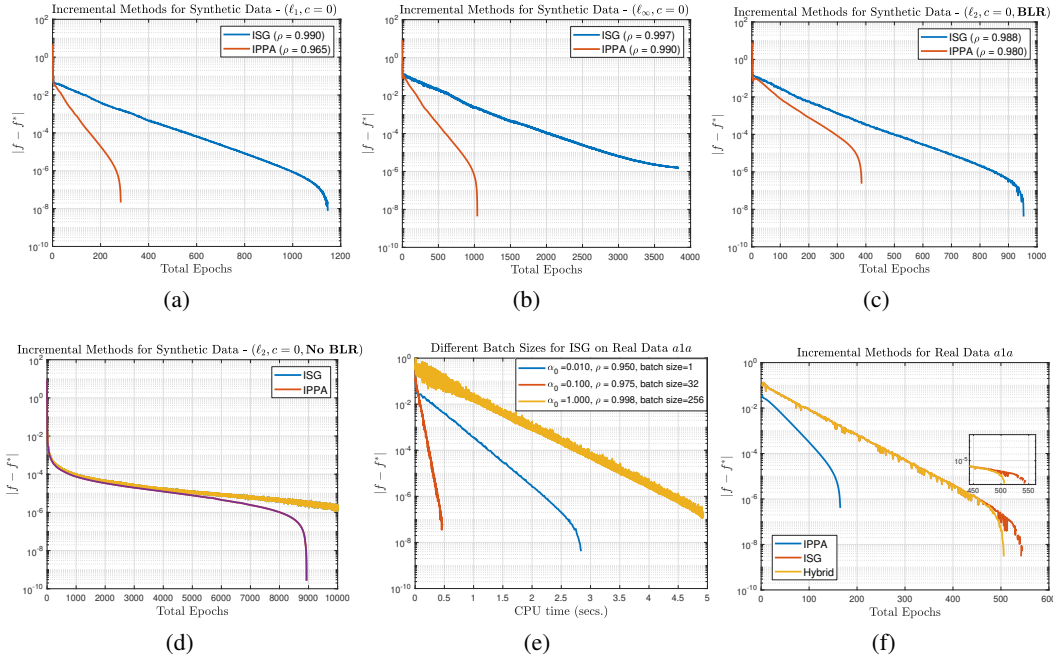

Figure 1: (a)–(d): Comparison between ISG and IPPA on both BLR and non-BLR instances generated from synthetic datasets. (e)–(f): Performance of ISG on different mini-batch sizes and performance of the hybrid algorithm on the `a1a` dataset.

compute the ground-truth labels $\{y_i\}_{i=1}^n$ by $y_i = \mathrm{sign}(\langle w^*, x_i \rangle + \xi_i)$. Here, the model parameters are $n = 1000, d = 100, \kappa = 1, \epsilon = 0.1$. All the algorithmic parameters of ISG and IPPA have been fine-tuned via grid search for optimal performance. Recall from Theorem 4.3 that for instances satisfying the sharpness condition, the smaller shrinking rate $\rho$ the algorithm can adopt, the faster its linear rate of convergence. The experiments results in Fig. 1(a–c) indicate that IPPA allows us to choose a more aggressive $\rho$ when compared with ISG over all instances satisfying the sharpness condition. A similar phenomenon has also been observed in previous works; see, e.g., [16, Fig. 1]. Even for instances that do not satisfy the sharpness condition, IPPA performs better than ISG; see Fig. 1(d).

Nevertheless, IPPA can only handle one sample at a time. Thus, we are motivated to develop an approach that can combine the best features of both ISG and IPPA. Towards that end, observe from Fig.1(e) that there is a tradeoff between the mini-batch size and the shrinking rate $\rho$, which means that there is an optimal mini-batch size for achieving the fastest convergence speed. Inspired by this, we propose to first apply the mini-batch ISG to obtain an initial point and then use IPPA in a local region around the optimal point to gain further speedup and get a more accurate solution. As shown in Fig. 1(f), such a hybrid algorithm is effective, thus confirming our intuition.

## 5.2 Efficiency of our incremental algorithms

Next, we demonstrate the efficiency of our proposed methods on the real datasets `a1a-a9a,ijcnn1` downloaded from the LIBSVM[2]. The results for $\ell_1$-DRSVM, which satisfies the sharpness condition, are shown in Table 2. Apparently, IPPA is slower than mini-batch ISG (i.e., M-ISG) in general but can obtain more accurate solutions. More importantly, the hybrid algorithm, which combines the advantages of both M-ISG and IPPA, has an excellent performance and achieves a well-balanced tradeoff between accuracy and efficiency. All of them are much faster than YALMIP. The results for $\ell_2$-DRSVM are reported in Table 3. As ISG is sensitive to hyper-parameters and has difficulty achieving the desired accuracy, we only present the results for IPPA. From the table, the superiority of IPPA over the solver is obvious.

Table 2: Wall-clock Time Comparison on UCI Real Dataset: $\ell_1$-DRSVM, $c = 0, \kappa = 1, \epsilon = 0.1$

| | Objective Value | | | | Wall-clock time (sec) | | | |
|---|---|---|---|---|---|---|---|---|
| | M-ISG | IPPA | Hybrid | YALMIP | M-ISG | IPPA | Hybrid | YALMIP |
| a1a | **0.651090** | 0.651091 | **0.651090** | 0.651102 | **0.706** | 6.1242 | 1.560 | 12.221 |
| a2a | **0.670640** | 0.670640 | **0.670640** | 0.670652 | **0.717** | 7.040 | 1.720 | 9.695 |
| a3a | 0.662962 | 0.663093 | **0.662962** | 0.663060 | **1.800** | 21.242 | 3.740 | 11.854 |
| a4a | 0.674274 | 0.674274 | **0.674273** | 0.674274 | **3.764** | 25.980 | 4.664 | 16.638 |
| a5a | **0.660867** | **0.660867** | **0.660867** | 0.660869 | **2.026** | 24.752 | 24.752 | 24.207 |
| a6a | **0.654189** | **0.654189** | **0.654189** | 0.654194 | **2.277** | 26.127 | 2.509 | 39.311 |
| a7a | 0.656274 | 0.656274 | **0.656273** | 0.656411 | **2.528** | 33.094 | 2.799 | 60.046 |
| a8a | 0.650036 | 0.650036 | **0.650035** | 0.650081 | **3.004** | 41.249 | 3.729 | 94.377 |
| a9a | 0.642186 | 0.642186 | **0.642185** | 0.642596 | **2.285** | 35.554 | 3.063 | 155.980 |

Table 3: Wall-clock Time Comparison on UCI Real Dataset: $\ell_2$-DRSVM, $c = 0, \kappa = 1, \epsilon = 0.1$

| | Objective Value | | Wall-clock time (sec) | | Regularity Condition |
|---|---|---|---|---|---|
| | IPPA | YALMIP | IPPA | YALMIP | |
| a1a | 0.6339472 | **0.6338819** | **5.517** | 8.557 | Not Known |
| a2a | 0.6599856 | **0.6599099** | **9.355** | 11.989 | Not Known |
| a3a | 0.6443777 | **0.6442762** | **7.096** | 15.335 | Not Known |
| a4a | 0.6513987 | **0.6513899** | **14.162** | 23.122 | Not Known |
| a5a | 0.6484421 | **0.6484147** | **10.515** | 32.663 | Not Known |
| a6a | 0.6428831 | **0.6428806** | **15.195** | 67.695 | Not Known |
| a7a | **0.6459271** | 0.6462302 | **6.454** | 118.740 | Not Known |
| a8a | **0.6441057** | 0.6441057 | **27.242** | 161.000 | Not Known |
| a9a | **0.6389162** | 0.6437767 | **13.129** | 215.387 | Sharpness |
| ijcnn | **0.4781876** | 0.4781897 | **20.567** | 379.943 | Sharpness |

To further demonstrate the efficiency of our proposed hybrid algorithm, we compare it with GS-ADMM [15] and YALMIP on $\ell_\infty$-DRSVM, which again satisfies the sharpness condition. The results are shown in Table 4. The overall performance of our hybrid method dominates both GS-ADMM and YALMIP. Due to space limitation, we only present the results for the case $q \in \{1, 2, \infty\}$, $c = 0$. More numerical results can be found in the Appendix.

Table 4: Wall-clock Time Comparison on UCI Real Dataset: $\ell_\infty$-DRSVM, $c = 0, \kappa = 1, \epsilon = 0.1$

| | Hybrid | GS-ADMM | YALMIP | | Hybrid | GS-ADMM | YALMIP |
|---|---|---|---|---|---|---|---|
| a1a | **4.789** | 5.939 | 7.832 | a6a | **8.273** | 8.273 | 42.714 |
| a2a | **5.098** | 7.069 | 9.100 | a7a | **6.115** | 6.115 | 60.743 |
| a3a | 16.252 | **9.638** | 11.375 | a8a | **11.065** | 11.065 | 99.355 |
| a4a | **5.498** | 10.446 | 17.542 | a9a | **5.717** | 5.717 | 172.07 |
| a5a | **7.363** | 13.993 | 22.969 | ijcnn | **4.301** | 4.301 | 319.379 |

# 6   Conclusion and Future Work

In this paper, we developed two new and highly efficient epigraphical projection-based incremental algorithms to solve the Wasserstein DRSVM problem with $\ell_p$ norm-induced transport cost ($p \in \{1, 2, \infty\}$) and established their convergence rates. A natural future direction is to develop a mini-batch version of IPPA and extend our algorithms to the asynchronous decentralized parallel setting. Inspired by our paper, it would also be interesting to develop some new incremental/stochastic algorithms to tackle more general Wasserstein DRO problems; see, e.g., problem (11) in [19].

**Acknowledgment**  Caihua Chen is supported in part by the National Natural Science Foundation of China (NSFC) projects 71732003, 11871269 and in part by the Natural Science Foundation of Jiangsu Province project BK20181259. Anthony Man-Cho So is supported in part by the CUHK Research Sustainability of Major RGC Funding Schemes project 3133236.

**Broader Impact**  This work does not present any foreseeable societal consequence. A broader impact discussion is not applicable.

## Footnotes

[2] https://www.csie.ntu.edu.tw/~cjlin/libsvmtools/datasets/binary.html

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
