[Supplementary Material]

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

# Appendix

This supplementary document is the appendix section of the paper titled "**Fast Epigraphical Projection-based Incremental Algorithms for Wasserstein Distributionally Robust Support Vector Machine**". It is organized as follows. In Section A, we give the details of the algorithms for solving the subproblems (i.e., $\ell_q$ norm epigraphical projection and single-sample proximal point update (6)). In Section B, we prove the results in the section "Convergence Rate Analysis of Incremental Algorithms". In Section C, we describe how to extend the algorithm in [15] (i.e., GS-ADMM) to tackle our $\ell_\infty$-DRSVM problem. Subsequently, we provide additional experimental results to further demonstrate the effectiveness of our proposed method.

## A: Algorithmic ingredients in ISG and IPPA

To begin, we provide a summary in Table 5, which aims to help the reader find the related algorithmic details as quickly as possible.

Table 5: Summary of all ingredients in ISG and IPPA

| Cases | ISG epigraphical projection (5) | IPPA single sample update (6) |
|---|---|---|
| $\ell_2$ | closed-form; see Prop. 6.1 | exhaust all seven cases; see Table 6 analytic form for subcases; see Prop. 6.2, 6.4 |
| $\ell_1$ | quick-select algorithm in linear time | exhaust all five cases; see Alg. 2 |
| $\ell_\infty$ | Moreau's decomposition based on $\ell_1$ case | modified secant alg. 1 |

**Subproblems for $q = 2$.** It is well known that the $\ell_2$ norm epigraphic projection has a closed-form formula, which is given as follows:

**Proposition 6.1 (Adopted from [1, Theorem 3.3.6])** *Let* $L_2^d = \{(x, s) \in \mathbb{R}^d \times \mathbb{R} : \|x\|_2 \leq s\}$. *For any* $(x, t) \in \mathbb{R}^d \times \mathbb{R}$, *we have*

$$\text{proj}_{L_2^d}(x, s) = \begin{cases} \left( \frac{\|x\|_2 + s}{2\|x\|_2}, \frac{\|x\|_2 + s}{2} \right) & \|x\|_2 \geq |s|, \\ (0, 0) & s < \|x\|_2 < -s, \\ (x, s) & \|x\|_2 \leq s. \end{cases} \tag{15}$$

Recall that the $\ell_2$ single-sample proximal point subproblem (7) takes the form

$$\min_{w, \lambda} \underbrace{\max \left\{ 1 - w^T z_i, 1 + w^T z_i - \lambda \kappa, 0 \right\}}_{h_i(w, \lambda)} + \frac{1}{2\alpha}(\|w - \bar{w}\|_2^2 + (\lambda - \bar{\lambda})^2), \quad \text{s.t. } \|w\|_2 \leq \lambda.$$

We start with $\Gamma = 1, j = 1$. Problem (8) can be written as

$$\min_{w, \lambda} \frac{1}{2\alpha}(\|w - \bar{w} - \alpha z_i\|_2^2 + (\lambda - \bar{\lambda})^2), \quad \text{s.t. } \|w\|_2 \leq \lambda,$$

whose optimal solution is given by $(\hat{w}, \hat{\lambda}) = \text{proj}_{L_2^d}(\bar{w} + \alpha z_i, \bar{\lambda})$. We further check whether $(\hat{w}, \hat{\lambda})$ satisfies the optimality condition of problem (7); i.e., $\{(w, \lambda) : h_{i,1}(w, \lambda) > h_{i,2}(w, \lambda), h_{i,1}(w, \lambda) > h_{i,3}(w, \lambda)\} = \{(w, \lambda) : w^T z < \min(\frac{\lambda \kappa}{2}, 1)\}$. The other two cases (i.e., $\Gamma = 1, j = 2, 3$) follow the same procedure. Then, we proceed to consider $\Gamma = 2$ (e.g., $(j, j') = (1, 2)$). Problem (9) is equivalent to

$$\min_{w, \lambda} \frac{1}{2\alpha}(\|w - \bar{w} - \alpha z_i\|_2^2 + (\lambda - \bar{\lambda})^2), \quad \text{s.t. } w^T z_i = \frac{\kappa}{2}\lambda, \|w\|_2 \leq \lambda.$$

We now derive an analytic solution for its prototypical form (16) in Proposition 6.2, which also covers the other two cases (i.e., $\Gamma = 2, (j, j') = (1, 3), (2, 3)$).

**Proposition 6.2** *Given $\bar{w} \in \mathbb{R}^d$ and $\bar{\lambda}, a, b \in \mathbb{R}$, consider the following optimization problem:*

$$\begin{aligned}
\min_{w,\lambda} \quad & \frac{1}{2}\|w - \bar{w}\|_2^2 + \frac{1}{2}(\lambda - \bar{\lambda})^2 \\
\text{s.t.} \quad & w^T z_i = a\lambda + b \leftarrow \mu_1, \\
& \|w\|_2 \leq \lambda \leftarrow \mu_2,
\end{aligned} \tag{16}$$

*where $\mu_1$ and $\mu_2$ are the associated dual multipliers. Then, the optimal solution $(w^*, \lambda^*)$ to (16) is*

$$\text{PPA\_sub}(\bar{w}, \bar{\lambda}, a, b) \triangleq \left( \frac{\bar{w} - \mu_1^* z_i}{1 + 2\mu_2^*}, \frac{\bar{\lambda} + a\mu_1^*}{1 - 2\mu_2^*} \right),$$

*where $\mu_1^*$ and $\mu_2^*$ are the optimal dual multipliers. In particular, we have*

$$\mu_1^* = \frac{(1 - 2\mu_2^*)\bar{w}^T z_i - a(1 + 2\mu_2^*)\bar{\lambda} - b(1 - 2\mu_2^*)(1 + 2\mu_2^*)}{(1 + 2\mu_2^*)a^2 + (1 - 2\mu_2^*)\|z_i\|_2^2},$$

$$\mu_2^* = \begin{cases} 0, & \text{if } \|\bar{w} - \mu_1^* z_i\|_2 \leq \bar{\lambda} + a\mu_1^*, \\ \hat{\mu}_2, & \text{otherwise} \end{cases}$$

*and $\hat{\mu}_2$ is the root of the following **quartic equation** satisfying $\hat{\mu}_2 > 0$ and $\lambda^* \geq 0$:*

$$p_1 \mu_2^4 + p_2 \mu_2^3 + p_3 \mu_2^2 + p_4 \mu_2 + p_5 = 0.$$

*Here, $A = \|\bar{w}\|_2^2$, $B = \|z_i\|_2^2$, $C = \bar{w}^T z_i$, and*

$$\begin{aligned}
p_1 =& a^2 b^2 - Bb^2, \\
p_2 =& 4a^2 b^2, \\
p_3 =& -4Bab\bar{\lambda} + 2BCa\bar{\lambda} - 2Ca^3\bar{\lambda} - 4Ca^2 b + 2ABa^2 + 6a^2 b^2 \\
& + B^2\bar{\lambda}^2 + 2Bb^2 + BC^2 - Ba^2\bar{\lambda}^2 - C^2 a^2 - Aa^4 - AB^2, \\
p_4 =& -8Bab\bar{\lambda} + 4BCa\bar{\lambda} - 2Ba^2\bar{\lambda}^2 - 4Ca^3\bar{\lambda} - 8Ca^2 b - 2Aa^4 \\
& - 2BC^2 + 2AB^2 + 4a^2 b^2 + 2B^2\bar{\lambda}^2 + 2C^2 a^2, \\
p_5 =& -4Bab\bar{\lambda} + 2BCa\bar{\lambda} - 2Ca^3\bar{\lambda} - 4Ca^2 b - 2ABa^2 + a^2 b^2 \\
& + B^2\bar{\lambda}^2 + 3C^2 a^2 + BC^2 - Ba^2\bar{\lambda}^2 - Bb^2 - Aa^4 - AB^2.
\end{aligned} \tag{17}$$

**Proof** The Karush-Kuhn-Tucker (KKT) conditions of (16) are given by

$$\begin{cases}
w^* - \bar{w} + \mu_1^* z_i + 2\mu_2^* w^* = 0, \\
\lambda^* - \bar{\lambda} - a\mu_1^* - 2\mu_2^* \lambda^* = 0, \\
w^{*T} z_i = a\lambda^* + b, \\
\|w^*\|_2 \leq \lambda^*, \\
\mu_2^*(\|w^*\|_2^2 - \lambda^{*2}) = 0, \\
\mu_2^* \geq 0.
\end{cases} \tag{18}$$

Based on (18), we have

$$(1 + 2\mu_2^*)w^{*T} z_i - \bar{w}^T z_i + \mu_1^* \|z_i\|_2^2 = 0 \quad \Rightarrow \quad w^{*T} z_i = \frac{\bar{w}^T z_i - \mu_1^* \|z_i\|_2^2}{1 + 2\mu_2^*},$$

$$(1 - 2\mu_2^*)\lambda^* - \bar{\lambda} - a\mu_1^* = 0 \quad \Rightarrow \quad (1 - 2\mu_2^*)(a\lambda^* + b) = a\bar{\lambda} + a^2\mu_1^* + b(1 - 2\mu_2^*).$$

Plugging in $w^{*T} z_i = a\lambda + b$ yields

$$(1 - 2\mu_2^*)\frac{\bar{w}^T z_i - \mu_1^* \|z_i\|_2^2}{1 + 2\mu_2^*} = a\bar{\lambda} + a^2\mu_1^* + b(1 - 2\mu_2^*)$$

$$\Rightarrow \quad (1 - 2\mu_2^*)(\bar{w}^T z_i - \mu_1^* \|z_i\|_2^2) = a\bar{\lambda}(1 + 2\mu_2^*) + a^2\mu_1^*(1 + 2\mu_2^*)) + b(1 - 2\mu_2^*)(1 + 2\mu_2^*).$$

Then,

$$\mu_1^* = \frac{(1 - 2\mu_2^*)\bar{w}^T z_i - a(1 + 2\mu_2^*)\bar{\lambda} - b(1 - 2\mu_2^*)(1 + 2\mu_2^*)}{(1 + 2\mu_2^*)a^2 + (1 - 2\mu_2^*)\|z_i\|_2^2}. \tag{19}$$

To handle the complementary slackness condition $\mu_2^*(\|w^*\|_2^2 - \lambda^{*2}) = 0$, we consider the following two cases:

- Case 1: If $\mu_2^* = 0$, then we have $\mu_1^* = \frac{\bar{w}^T z_i - a\bar{\lambda} - b}{a^2 + \|z_i\|_2^2}$ and hence

$$w^* = \bar{w} - \mu_1^* z_i, \quad \lambda^* = \bar{\lambda} + a\mu_1^*.$$

  If $\|w^*\|_2 \leq \lambda^*$ does not hold, we go to Case 2.

- Case 2: If $\mu_2^* > 0$, then by incorporating $\|w^*\|_2 = \lambda^*$ into (19), we obtain the quartic equation

$$p_1\mu_2^4 + p_2\mu_2^3 + p_3\mu_2^2 + p_4\mu_2 + p_5 = 0,$$

  whose coefficients $p_1, \ldots, p_5$ are given in (17). Finally, $\mu_2^*$ is in effect the root of this quartic equation, which satisfies $\mu_2^* > 0$ and $\lambda^* \geq 0$. The optimal solution $(w^*, \lambda^*)$ is then given by

$$w^* = \frac{\bar{w} - \mu_1^* z_i}{1 + 2\mu_2^*}, \quad \lambda^* = \frac{\bar{\lambda} + a\mu_1^*}{1 - 2\mu_2^*}.$$

<div align="right">□</div>

**Remark 6.3** *The KKT conditions are necessary and sufficient for optimality for problem* (16)*. If there does not exist a KKT point* $(w^*, \lambda^*, \mu_1^*, \mu_2^*)$ *(i.e., no nonnegative roots for the quartic function), then the case* $\Gamma = 2$ *is not optimal for* (7) *and we proceed to other cases. For practical implementation, we apply the built-in function* `roots([p1,p2,p3,p4,p5])` *in MATLAB to get the roots of the quartic equation.*

Similarly, we check the corresponding optimality condition $\{(w, \lambda) : h_{i,1}(w, \lambda) = h_{i,2}(w, \lambda) > h_{i,3}(w, \lambda)\} = \{(w, \lambda) : \lambda\kappa - 1 < w^T z_i < 1, \lambda\kappa < 2\}$. The other two cases follow the same procedure. Lastly, we proceed to the case $\Gamma = 3$ and problem (9) in effect admits a closed-form update.

**Proposition 6.4** *Given* $\bar{w} \in \mathbb{R}^d$ *and* $b, \lambda \in \mathbb{R}$*, consider the following optimization problem:*

$$\begin{aligned} \min_{w} \quad & \frac{1}{2}\|w - \bar{w}\|_2^2 \\ \text{s.t.} \quad & w^T z_i = b \leftarrow \alpha, \\ & \|w\|_2 \leq \lambda \leftarrow \beta, \end{aligned} \tag{20}$$

*where* $\alpha$ *and* $\beta$ *are the associated dual multipliers. Then, the optimal solution* $w^*$ *to* (20) *is*

$$\mathcal{O}_{BH}(b, \lambda, \bar{w}) \triangleq \begin{cases} A, & \text{if } \|A\|_2 \leq \lambda, \\ \frac{1}{2\beta^* + 1}\left\{ A + \frac{2b\beta^*}{\|z_i\|_2^2} z_i \right\}, & \text{otherwise}, \end{cases}$$

*where* $A = \bar{w} - \frac{\bar{w}^T z_i - b}{\|z_i\|_2^2} z_i$*,* $B = \frac{2b}{\|z_i\|_2^2} z_i$*, and* $\beta^*$ *is the positive root of the following **quadratic** equation:*

$$(4\lambda^2 - \|B\|_2^2)\beta^2 + (4\lambda^2 - 2A^T B)\beta + (\lambda^2 - \|A\|_2^2) = 0.$$

**Proof** The KKT conditions of (20) are given by

$$\begin{cases} w^* - \bar{w} + \alpha^* z_i + 2\beta^* w^* = 0, \\ w^{*T} z_i - b = 0, \\ \|w^*\|_2 \leq \lambda, \\ \beta^*(\|w^*\|_2^2 - \lambda^2) = 0, \\ \beta^* \geq 0. \end{cases} \tag{21}$$

Here, $\alpha^*$ and $\beta^*$ are the optimal dual multipliers. On top of (21), we have

$$(1 + 2\beta^*)w^* - \bar{w} + \alpha^* z_i = 0 \quad \Rightarrow \quad (1 + 2\beta^*)w^{*T} z_i - \bar{w}^T z_i + \alpha^*\|z_i\|_2^2 = 0,$$

$$(1 + 2\beta^*)b - \bar{w}^T z_i + \alpha^*\|z_i\|_2^2 = 0 \quad \Rightarrow \quad \alpha^* = \frac{\bar{w}^T z_i - (1 + 2\beta^*)b}{\|z_i\|_2^2}.$$

Plugging in $w^* = \frac{\bar{w} - \alpha^* z_i}{1 + 2\beta^*}$ gives

$$w^* = \frac{1}{2\beta^* + 1} \left\{ \bar{w} - \frac{\bar{w}^T z - (1 + 2\beta^*)b}{\|z_i\|_2^2} z_i \right\}.$$

Similarly, to handle the complementary slackness condition, we consider the case $\beta^* = 0$. For this case, we check whether the condition $\|A\|_2 \leq \lambda$ holds. Otherwise, $\beta^* > 0$ and $\|w^*\|_2^2 = \lambda^2$. This is equivalent to finding the positive root of the quadratic function

$$(4\lambda^2 - \|B\|_2^2)\beta^{*2} + (4\lambda^2 - 2A^T B)\beta^* + (\lambda^2 - \|A\|_2^2) = 0.$$

The geometric interpretation of this case is illustrated in Fig. 2. Problem (20) seeks to find the projection onto the intersection of the Euclidean ball $\|w\|_2 \leq \lambda$ and the hyperplane $w^T z_i = b$. Observe that the projection of $\bar{w}$ onto the hyperplane $w^T z_i = b$ is given by $A = \bar{w} - \frac{\bar{w}^T z_i - b}{\|z_i\|_2^2} z_i$. If $A \in \{w : \|w\|_2 \leq \lambda\}$ (i.e., red case), then $A$ is the optimal solution to (20). Otherwise, we aim to find the point on the sphere (i.e., $\|w^*\|_2 = \lambda$) that is closer to $A$ (i.e., blue case).

Figure 2: Projection onto the intersection of Euclidean ball and hyperplane.

$\square$

Let us now summarize the optimal solutions of all sub-cases and the corresponding optimality conditions in Table 6.

Table 6: Summary of all sub-cases for $\ell_2$ proximal point update (7)

| Sub-cases | Optimal solution | Optimality condition |
|---|---|---|
| $\Gamma = 1, j = 1$ | $\mathrm{proj}_{L_2^d}(\bar{w} + \alpha z_i, \bar{\lambda})$ | $w^{*T} z < \min(\frac{\lambda^* \kappa}{2}, 1)$ |
| $\Gamma = 1, j = 2$ | $\mathrm{proj}_{L_2^d}(\bar{w} - \alpha z_i, \bar{\lambda} + \alpha \kappa)$ | $w^{*T} z > \max(\frac{\lambda^* \kappa}{2}, \lambda^* \kappa - 1)$ |
| $\Gamma = 1, j = 3$ | $\mathrm{proj}_{L_2^d}(\bar{w}, \bar{\lambda})$ | $1 < w^{*T} z < \lambda^* \kappa - 1$ & $\lambda^* \kappa > 2$ |
| $\Gamma = 2, (j, j') = (1, 2)$ | $\mathrm{PPA\_sub}(\bar{w} + \alpha z_i, \bar{\lambda}, \frac{\kappa}{2}, 0)$ | $\lambda^* \kappa - 1 < w^{*T} z < 1$ & $\lambda^* \kappa < 2$ |
| $\Gamma = 2, (j, j') = (1, 3)$ | $\mathrm{PPA\_sub}(\bar{w}, \bar{\lambda}, 0, 1)$ | $w^{*T} z < \min(\frac{\lambda^* \kappa}{2}, \lambda^* \kappa - 1)$ & $\lambda^* \kappa > 2$ |
| $\Gamma = 2, (j, j') = (2, 3)$ | $\mathrm{PPA\_sub}(\bar{w}, \bar{\lambda}, \kappa, -1)$ | $w^{*T} z > \max(\frac{\lambda^* \kappa}{2}, 1)$ & $\lambda^* \kappa > 2$ |
| $\Gamma = 3$ | $(\mathcal{O}_{BH}(1, \frac{2}{\kappa}, \bar{w}), \frac{2}{\kappa})$ | $\lambda^* = \frac{2}{\kappa}$ |

**Subproblems for $q = 1$.** Consider the $\ell_1$ norm epigraphical projection

$$\mathrm{proj}_{L_1^d}(x, s) = \arg\min_{y, t} \left\{ \frac{1}{2}\|y - x\|_2^2 + \frac{1}{2}(t - s)^2, \text{ s.t. } \|y\|_1 \leq t \right\}, \tag{22}$$

where $L_1^d = \{(x, s) \in \mathbb{R}^d \times \mathbb{R} : \|x\|_1 \leq s\}$. Problem (22) is equivalent to finding the root of an one-dimensional piecewise linear equation. By inspecting the KKT conditions (with $\lambda \geq 0$

being the Lagrangian multiplier), we have $y = \text{sign}(x) \odot \max(|x| - \lambda, 0)$ (i.e., proximal operator for $\ell_1$ norm) and $t = \lambda + s$. Combining this with the complementary slackness condition and $\lambda > 0$, the KKT conditions of (22) reduce to the piecewise linear root-finding problem $F(\lambda) = \sum_{i=1}^{d} \max(|x_i| - \lambda) - \lambda - s = 0$, which can be solved by the quick-select algorithm in linear time; see [26, Algorithm 2] for details. Otherwise, we have $\text{proj}_{L_1^d}(x, s) = (x, s)$.

Recall that the $\ell_1$ single-sample proximal point subproblem (11) takes the form

$$\min_{w, \lambda, \mu} \mu + \frac{1}{2\alpha} \left( \|w - \bar{w}\|_2^2 + (\lambda - \bar{\lambda})^2 \right)$$
$$\text{s.t.} \quad h_{i,j}(w, \lambda) \le \mu \ (\leftarrow \sigma_j \ge 0), \ j = 1, 2, 3; \ \|w\|_1 \le \lambda,$$

where $\sigma_1, \sigma_2, \sigma_3 \ge 0$ are the corresponding dual multipliers.

- Case 1: $h_{i,1}, h_{i,3}$ are inactive. Then, problem (11) can be written as

$$\min_{w, \lambda} (1 + w^T z_i - \lambda \kappa) + \frac{1}{2\alpha} \|w - \bar{w}\|_2^2 + \frac{1}{2\alpha} (\lambda - \bar{\lambda})^2, \ \text{s.t.} \ \|w\|_1 \le \lambda.$$

  Hence, we have $(w^*, \lambda^*) = \text{proj}_{L_1^d}(\bar{w} - \alpha z_i, \bar{\lambda} + \alpha \kappa)$.

- Case 2: $h_{i,1}$ is active and $h_{i,3}$ is inactive. Then, problem (12) can be reduced to

$$\min_{w, \lambda} \frac{1}{2\alpha}(\|w - \bar{w} - \alpha z_i\|_2^2 + (\lambda - \bar{\lambda})^2), \ \text{s.t.} \ w^T z_i \le \frac{\lambda \kappa}{2} \ (\leftarrow \sigma_1 \ge 0), \ \|w\|_1 \le \lambda. \quad (23)$$

- Cases 3 and 4: ($h_{i,1}$ is inactive, $h_{i,3}$ is active) and ($h_{i,2}$ is inactive, $h_{i,3}$ is active). These two cases are similar to Case 2 and give rise to a problem of the form (13).

Now, let us demonstrate how to solve (13) efficiently. Recall that

$$\min_{w, \lambda} \frac{1}{2\alpha}(\|w - \bar{w}\|_2^2 + (\lambda - \bar{\lambda})^2), \ \text{s.t.} \ w^T z \le a\lambda + b \ (\leftarrow \sigma \ge 0), \ \|w\|_1 \le \lambda.$$

**Proposition 6.5** *Suppose that $\sigma_1^*$ is the dual optimal solution to (23). Then, we have $\sigma_1^* \in [0, 1]$.*

**Proof** Based on the KKT conditions of (11), we have

$$1 - \sigma_1 - \sigma_2 - \sigma_3 = 0.$$

If the optimal solution to (23) is also optimal for (11), then we can match the two KKT systems. As $h_{i,3}$ is inactive for this case, we have $\sigma_3^* = 0$. This gives

$$\sigma_1^* + \sigma_2^* = 1, \sigma_1^*, \sigma_2^* \ge 0 \quad \Rightarrow \quad \sigma_1^* \in [0, 1].$$

$\square$

Proposition 6.5 also holds for Cases 3 and 4. The analytic bound in Proposition 6.5 shows that $\sigma^*$ can be efficiently found by an appropriate search strategy. Next, recall from (14) that

$$(\hat{w}(\sigma), \hat{\lambda}(\sigma)) = \arg\min_{\|w\|_1 \le \lambda} \frac{1}{2\alpha} \left( \|w - \bar{w}\|_2^2 + (\lambda - \bar{\lambda})^2 \right) + \sigma(w^T z - a\lambda - b)$$
$$= \text{proj}_{L_1^d}(\bar{w} - \sigma\alpha z, \bar{\lambda} + \sigma\alpha a).$$

The following proposition establishes the monotonicity property of $\sigma \mapsto p(\sigma) = \hat{w}(\sigma)^T z - a\kappa - b$, which plays a vital role in our development of a fast algorithm for solving (13) later.

**Proposition 6.6** *If $\sigma$ satisfies (i) $\sigma = 0$ and $p(\sigma) \le 0$, or (ii) $p(\sigma) = 0$, then $(\hat{w}(\sigma), \hat{\lambda}(\sigma))$ is the optimal solution to (13). Moreover, $p(\cdot)$ is continuous and monotonically non-increasing on $\mathbb{R}_+$.*

**Proof** As $\text{proj}_{L_1^d}(\cdot, \cdot)$ is globally Lipschitz continuous, the function $(\hat{w}(\cdot), \hat{\lambda}(\cdot))$ is also globally Lipschitz continuous and further $p(\cdot)$ is continuous. Next, we prove the monotonicity property. Upon letting $h(\sigma) = \frac{1}{2\alpha}\left(\|\hat{w}(\sigma) - \bar{w}\|_2^2 + (\hat{\lambda}(\sigma) - \bar{\lambda})^2\right)$ and assuming that $0 \le \sigma_1 < \sigma_2 \le 1$, we have

$$
\begin{aligned}
h(\sigma_1) + \sigma_1 p(\sigma_1) &\le h(\sigma_2) + \sigma_1 p(\sigma_2) \\
&= h(\sigma_2) + \sigma_2 p(\sigma_2) + (\sigma_1 - \sigma_2)p(\sigma_2) \\
&\le h(\sigma_1) + \sigma_2 p(\sigma_1) + (\sigma_1 - \sigma_2)p(\sigma_2),
\end{aligned}
$$

which implies that $p(\sigma_1) \ge p(\sigma_2)$. $\qquad\square$

---

**Algorithm 1:** A modified secant algorithm to solve the RHS of (13)—MSA($\bar{w}, \bar{\lambda}, z_i, a, b, \xi$)

---

**Input:** Tolerance Level $\xi$ ;
**if** $p(0) \le \xi$ **then** return $(\hat{w}(0), \hat{\lambda}(0), 0)$ ;
**else**

    $\sigma_l = 0, r_l = -p(0)$ ;   // Set the lower bound $\sigma_l$ for $\sigma^*$
    **if** $p(1) \ge 0$ **then** return $-1$ ;
    // $\sigma^* \in [0,1]$; see Proposition 3.1
    **else**
       |  $\sigma_u = 1, r_u = -p(1)$ ;   // Set the upper bound $\sigma_u$ for $\sigma^*$
    **end**
**end**
/* Secant Phase                                                            */
$s = 1 - \frac{r_l}{r_u}$, $\sigma = \sigma_u - \frac{\sigma_u - \sigma_l}{s}$; calculate $r = -p(\sigma)$ ;
**while** $|r| > \xi$ **do**
    calculate $r = -p(\sigma)$;   // $\ell_1$ epigraph projection via the quick-select algorithm
    **if** $r > 0$ **then**
        **if** $s \le 2$ **then** $\sigma_u = \sigma$, $r_u = r$, $s = 1 - \frac{r_L}{r_u}$, $\sigma = \sigma_u - \frac{\sigma_u - \sigma_l}{s}$ ;
        **else**
            |  $s = \max(\frac{r_u}{r} - 1, 0.1)$, $\Delta\sigma = \frac{\sigma_u - \sigma}{s}$, $\sigma_u = \sigma$, $r_u = r$ ;
            |  $\sigma = \max(\sigma_u - \Delta\sigma, 0.6\sigma_l + 0.4\sigma_u)$, $s = \frac{\sigma_u - \sigma_l}{\sigma_u - \sigma}$ ;
        **end**
    **else**
        **if** $s \ge 2$ **then** $\sigma_l = \sigma$, $r_l = r$, $s = 1 - \frac{r_l}{r_u}$, $\sigma = \sigma_u - \frac{\sigma_u - \sigma_l}{s}$;
        **else**
            |  $s = \max(\frac{r_l}{r} - 1, 0.1)$, $\Delta\sigma = \frac{\sigma - \sigma_l}{s}$, $\sigma_l = \sigma$, $r_l = r$ ;
            |  $\sigma = \max(\sigma_l + \Delta\sigma, 0.6\sigma_u + 0.4\sigma_l)$, $s = \frac{\sigma_u - \sigma_l}{\sigma_u - \sigma}$ ;
        **end**
    **end**
**end**

---

## B: Convergence Rate Analysis of Incremental Algorithms

We now give a condition under which problem (1) with $q = 2$ satisfies the sharpness or quadratic growth (QG) property. Consider the following more general formulation of the $\ell_2$-DRSVM problem:

$$
\min_{w, \lambda} \frac{c}{2}\|w\|_2^2 + \frac{1}{n}\sum_{i=1}^n f_i(w, \lambda) + \mathbb{I}_{\{(w, \lambda) \in L_2^d\}}, \tag{24}
$$

where $f_1, \ldots, f_n$ are non-smooth convex functions with polyhedral epigraphs. Our condition is based the following lemma:

---
**Algorithm 2:** A fast algorithm based on parametric approach to solve (11)
---
**Input:** Tolerance Level $\xi$; parameters $\bar{w}, \bar{\lambda}, z_i, \epsilon, \kappa$ ;

```
/* Case 1:  h_{i,1}, h_{i,3} are inactive                                  */
```
$(w^*, \lambda^*) = \text{proj}_{L_1^d}(\bar{w} - \alpha z_i, \bar{\lambda} + \alpha\kappa)$ ;

**if** $\langle w^*, z \rangle > \max(\lambda^*\kappa - 1, \frac{\lambda^*\kappa}{2})$ **then** return $(w^*, \lambda^*)$ ;

```
// Check optimality
/* Case 2:  h_{i,1} is active; h_{i,3} is inactive                         */
```
$(w^*, \lambda^*, \sigma^*) = \text{MSA}(\bar{w} + \alpha z_i, \bar{\lambda}, z_i, \kappa/2, 0)$ ; `// Apply the modified secant algorithm 1`

**if** $\langle w^*, z \rangle < 1$ `&` $\sigma^* \in [0, 1]$ **then** return $(w^*, \lambda^*)$ ;

```
/* Case 3: h_{i,1} is inactive; h_{i,3} is active                          */
```
$(w^*, \lambda^*, \sigma^*) = \text{MSA}(\bar{w}, \bar{\lambda}, z_i, \kappa, -1)$ ;

**if** $\langle w^*, z \rangle > 1$ `&` $\sigma^* \in [0, 1]$ **then** return $(w^*, \lambda^*)$ ;

```
/* Case 4:  h_{i,2} is inactive; h_{i,3} is active                         */
```
$(w^*, \lambda^*, \sigma^*) = \text{MSA}(\bar{w}, \bar{\lambda}, -z_i, 0, -1)$ ;

**if** $\lambda^*\kappa > 2$ `&` $\sigma^* \in [0, 1]$ **then** return $(w^*, \lambda^*)$ ;

```
/* Case 5:  h_{i,1}, h_{i,2}, h_{i,3} are active                           */
```
**else**

> $w^* = \underset{w}{\arg\min}\{\|w - \bar{w}\|_2^2, \text{ s.t. } w^T z_i = 1, \|w\|_1 \leq \frac{2}{\kappa}\}, \lambda^* = \frac{\kappa}{2}$ ;
>
> `/* Apply the modified secant algorithm in [9]                           */`

**end**
---

**Lemma 1** *Let $C_1, \ldots, C_N$ be closed convex subsets of $\mathbb{R}^n$, where $C_{r+1}, \ldots, C_N$ are polyhedral for some $r \in \{0, 1, \ldots, N\}$. Suppose that*

$$\bigcap_{i=1}^{r} \text{ri}(C_i) \cap \bigcap_{i=r+1}^{N} C_i \neq \emptyset.$$

*Then, the collection $\{C_1, \ldots, C_N\}$ is boundedly linearly regular (BLR).*

**Proposition 6.7** *Consider problem (24). Let $\mathcal{X}$ be the set of optimal solutions and $L_2^d = \{(w, \lambda) \in \mathbb{R}^d \times \mathbb{R} : \|w\|_2 \leq \lambda\}$ be the constraint set. Suppose that $\mathcal{X} \cap \text{ri}(L_2^d) \neq \emptyset$. Then, problem (1) satisfies the sharpness condition when $c = 0$ and the QG condition when $c > 0$.*

**Proof** Let $x = (w, \lambda)$, $h(x) = \frac{c}{2}\|w\|_2^2 + \frac{1}{n}\sum_{i=1}^{n} f_i(w, \lambda)$, and $g(x) = h(x) + \mathbb{I}_{\{x \in L_2^d\}}$. Consider the case where $c = 0$. The set $\mathcal{X}$ can then be written as

$$\mathcal{X} = \{x : 0 \in \partial h(x) + \mathcal{N}_{L_2^d}(x)\},$$

where $\mathcal{N}_{L_2^d}(x)$ is the normal cone of $L_2^d$ at $x$. As $\mathcal{X} \cap \text{ri}(L_2^d) \neq \emptyset$, we can find an $x^* \in \mathcal{X} \cap \text{ri}(L_2^d)$ that satisfies $0 \in \partial h(x^*)$. Thus, $x^*$ is also an optimal solution to the unconstrained problem $\min_x h(x)$ and $h(x^*) = g(x^*)$.

Let $\mathcal{X}_U$ denote the set of optimal solutions to the problem $\min_x h(x)$. It is not difficult to check that

$$\mathcal{X} = \mathcal{X}_U \cap L_2^d.$$

Since $f_1, \ldots, f_n$ have polyhedral epigraphs, by Lemma 1, the collection $\{\mathcal{X}_U, L_2^d\}$ is BLR. This implies that there exists a constant $\kappa > 0$ satisfying

$$\text{dist}(x, \mathcal{X}) = \text{dist}(x, \mathcal{X}_U \cap L_2^d) \leq \kappa \, \text{dist}(x, \mathcal{X}_U), \quad \forall x \in L_2^d.$$

Furthermore, the problem $\min_x h(x)$ enjoys the sharpness property; see [8, Corollary 3.6]. This gives

$$g(x) - g^* = h(x) - h^* \geq \sigma \, \text{dist}(x, \mathcal{X}_U) \geq \frac{\sigma}{\kappa} \, \text{dist}(x, \mathcal{X}), \quad \forall x \in L_2^d.$$

For $c > 0$, we note that the problem $\min_x h(x)$ can be regarded as one with a polyhedral convex regularizer; see [28, Section 4.2]. As such, it satisfies a proximal error bound (see [28, Proposition 6]) and hence the QG condition (see [14, Theorem 4.1]). It follows that

$$g(x) - g^* = h(x) - h^* \geq \sigma \operatorname{dist}^2(x, \mathcal{X}_U) \geq \frac{\sigma}{\kappa^2} \operatorname{dist}^2(x, \mathcal{X}), \ \ \forall x \in L_2^d.$$

$\square$

To derive the convergence rates of the incremental algorithms, we also need the following assumption.

**Assumption 6.8 (Subgradient boundedness)** *There exists a scalar $L > 0$ such that*

$$\|\nabla f_i(x)\| \leq L, \ \ \forall \nabla f_i(x) \in \partial f_i(x), \ i \in [n].$$

**Lemma 2 (ISG; see [21, Lemma 2.1])** *Suppose that Assumption 6.8 holds and $\{x^k = (w_0^k, \lambda_0^k)\}$ is a sequence generated by ISG. Then, for all $y$ and $k \geq 0$, we have*

$$\|x^{k+1} - y\|_2^2 \leq \|x^k - y\|_2^2 - 2\alpha_k n(f(x^k) - f(y)) + a_k^2 n^2 L^2.$$

**Lemma 3 (IPPA)** *Suppose that Assumption 6.8 holds and $\{x^k = (w_0^k, \lambda_0^k)\}$ is a sequence generated by IPPA. Then, for all $y$ and $k \geq 0$, we have*

$$\|x^{k+1} - y\|_2^2 \leq \|x^k - y\|_2^2 - 2\alpha_k n(f(x^k) - f(y)) + a_k^2 n(n+1)L^2.$$

**Proof** Based on Proposition 1 in [3] with $x_i^k = (w_i^k, \lambda_i^k)$, we have

$$\|x_{i+1}^k - y\|_2^2 \leq \|x_i^k - y\|_2^2 - 2\alpha_k(f_{i+1}(x_{i+1}^k) - f_{i+1}(y)), \ \ \forall i = 0, \dots, n-1.$$

Summing up,

$$
\begin{aligned}
\|x_n^k - y\|_2^2 &\leq \|x_0^k - y\|_2^2 - 2\alpha_k \sum_{i=0}^{n-1}(f_{i+1}(x_{i+1}^k) - f_{i+1}(y)) \\
&= \|x_0^k - y\|_2^2 - 2\alpha_k \sum_{i=0}^{n-1}(f_{i+1}(x_{i+1}^k) - f_{i+1}(x_0^k) + f_{i+1}(x_0^k) - f_{i+1}(y)) \\
&= \|x_0^k - y\|_2^2 - 2\alpha_k n(f(x_0^k) - f(y)) - 2\alpha_k \sum_{i=0}^{n-1}(f_{i+1}(x_{i+1}^k) - f_{i+1}(x_0^k)) \\
&\leq \|x_0^k - y\|_2^2 - 2\alpha_k n(f(x_0^k) - f(y)) + 2\alpha_k L \sum_{i=0}^{n-1} \|x_{i+1}^k - x_0^k\|_2 \\
&\leq \|x_0^k - y\|_2^2 - 2\alpha_k n(f(x_0^k) - f(y)) + 2\alpha_k^2 L^2 \sum_{i=0}^{n-1}(i+1) \\
&\leq \|x_0^k - y\|_2^2 - 2\alpha_k n(f(x_0^k) - f(y)) + a_k^2 n(n+1)L^2.
\end{aligned}
$$

$\square$

Combining Lemmas 2 and 3, we have

$$\|x^{k+1} - y\|_2^2 \leq \|x^k - y\|_2^2 - 2\alpha_k n(f(x^k) - f(y)) + 2a_k^2 n^2 L^2.$$

**Theorem 6.9** *Let $\{x^k = (w_0^k, \lambda_0^k)\}$ be the sequence of iterates generated by ISG or IPPA.*

*(1) If problem (1) satisfies the sharpness condition, then by choosing the geometrically diminishing step sizes $\alpha_k = \alpha_0 \rho^k$ with $\alpha_0 \geq \frac{\sigma \operatorname{dist}(x^0, \mathcal{X})}{2L^2 n}$ and $\sqrt{1 - \frac{\sigma^2}{2L^2}} \leq \rho < 1$, the sequence $\{x^k\}$ converges linearly to an optimal solution to (1); i.e., $\operatorname{dist}(x^k, \mathcal{X}) \leq \mathcal{O}(\rho^k)$ for all $k \geq 0$.*

(2) *If problem* (1) *satisfies the* quadratic growth *condition, then by choosing the polynomially decaying step sizes* $\alpha_k = \frac{\gamma}{nk}$ *with* $\gamma > \frac{1}{2\sigma}$, *the sequence* $\{x^k\}$ *converges to an optimal solution to* (1) *at the rate* $\mathcal{O}(\frac{1}{\sqrt{k}})$ *and* $\{f(x^k) - f^*\}$ *converges to zero at the rate* $\mathcal{O}(\frac{1}{k})$.

(3) *(See [20, Proposition 2.10]) For the general convex problem* (1)*, by choosing the step sizes* $\alpha_k = \frac{\gamma}{n\sqrt{k}}$ *with* $\gamma > 0$*, the sequence* $\{\min_{0 \le k \le K} f(x^k) - f^*\}$ *converges to zero at the rate* $\mathcal{O}(\frac{1}{\sqrt{K}})$.

**Proof** (1):
By the sharpness condition $f(x_k) - f^* \ge \sigma \operatorname{dist}(x_k, \mathcal{X})$, we have

$$\operatorname{dist}^2(x^{k+1}, \mathcal{X}) \le \operatorname{dist}^2(x^k, \mathcal{X}) - 2\alpha_k \sigma n \operatorname{dist}(x^k, \mathcal{X}) + 2\alpha_k^2 L^2 n^2.$$

We now prove by induction that

$$\operatorname{dist}(x^k, \mathcal{X}) \le \frac{2\alpha_0 L^2 n}{\sigma} \rho^k.$$

The base case trivially holds, as $\operatorname{dist}(x^0, \mathcal{X}) \le \frac{2\alpha_0 L^2 n}{\sigma}$. For the inductive step, we compute

$$
\begin{aligned}
\operatorname{dist}^2(x^{k+1}, \mathcal{X}) &\le \left( \frac{2\alpha_0 L^2 n}{\sigma} \rho^k \right)^2 - 2\alpha_k \sigma n \frac{2\alpha_0 L^2 n}{\sigma} \rho^k + 2\alpha_k^2 L^2 n^2 \\
&= \frac{4\alpha_0^2 L^4 n^2}{\sigma^2} \rho^{2k} - 2\alpha_0^2 L^2 n^2 \rho^{2k} \\
&= \frac{4\alpha_0^2 L^4 n^2}{\sigma^2} \rho^{2k} \left( 1 - \frac{\sigma^2}{2L^2} \right) \\
&\le \left( \frac{2\alpha_0 L^2 n}{\sigma} \right)^2 \rho^{2(k+1)}.
\end{aligned}
\tag{25}
$$

This completes the proof.

(2): By the quadratic growth condition $f(x_k) - f^* \ge \sigma \operatorname{dist}^2(x_k, \mathcal{X})$, we have

$$\operatorname{dist}^2(x^{k+1}, \mathcal{X}) \le (1 - 2\alpha_k \sigma n) \operatorname{dist}^2(x^k, \mathcal{X}) + 2\alpha_k^2 L^2 n^2.$$

Plugging in the corresponding step size scheme $\alpha_k = \frac{\gamma}{nk}$, we obtain

$$\operatorname{dist}^2(x^{k+1}, \mathcal{X}) \le (1 - \frac{2\gamma\sigma}{k}) \operatorname{dist}^2(x^k, \mathcal{X}) + \frac{2\gamma^2 L^2}{k^2}.$$

We now prove by induction that

$$\operatorname{dist}^2(x^k, \mathcal{X}) \le \frac{B}{k},$$

where $B > 0$ is a given number. Indeed, we have

$$
\begin{aligned}
\operatorname{dist}^2(x^{k+1}, \mathcal{X}) &\le \left( 1 - \frac{2\gamma\sigma}{k} \right) \frac{B}{k} + \frac{2\gamma^2 L^2}{k^2} \\
&= \frac{B}{k+1} + \frac{B}{k(k+1)} - \frac{2\gamma\sigma B}{k^2} + \frac{2\gamma^2 L^2}{k^2} \\
&\le \frac{B}{k+1} + \frac{B}{k^2} - \frac{2\gamma\sigma B}{k^2} + \frac{2\gamma^2 L^2}{k^2} \\
&= \frac{B}{k+1} + \frac{(1 - 2\gamma\sigma)B}{k^2} + \frac{2\gamma^2 L^2}{k^2} \\
&\le \frac{B}{k+1},
\end{aligned}
$$

where the last inequality holds if $\frac{(1-2\gamma\sigma)B}{k^2} + \frac{2\gamma^2 L^2}{k^2} < 0$. Hence, we have $B > \frac{2\gamma^2 L^2}{2\gamma\sigma-1}$ due to $\gamma > \frac{1}{2\sigma}$. Combining this with the base case, we have $B > \max\{\frac{2\gamma^2 L^2}{2\gamma\sigma-1}, \operatorname{dist}^2(x^0, \mathcal{X})\}$. $\qquad\square$

## C: Additional Experimental Results

To begin with, we show how to extend the GS-ADMM framework in [15] to tackle our DRSVM problems, which serves as a baseline in Section 5. We reformulate problem (1) as

$$
\begin{aligned}
\min_{w,s,\lambda} \quad & \lambda\epsilon + \frac{1}{n}\sum_{i=1}^{n} s_i \\
\text{s.t.} \quad & 1 - w^T z_i \leq s_i, i \in [n], \\
& 1 + w^T z_i + \lambda\kappa \leq s_i, i \in [n], \\
& s_i \geq 0, i \in [n], \\
& \|w\|_q \leq \lambda.
\end{aligned}
\tag{26}
$$

We follow the technique used in [15, Proposition 3.1].

**Proposition 6.10** *Suppose that* $(w^*, \lambda^*, s^*)$ *is a global minimizer of* (26). *Then, we have* $\lambda^* \leq \lambda^U = \frac{1}{\epsilon}$.

**Proof** For simplicity, we consider the case where $q = 2$. Since problem (26) satisfies the Managasarian-Fromovitz Constraint Qualification (MFCQ), the KKT conditions are necessary and sufficient. Let $a_{ij} \geq 0, \forall j \in [3], i \in [N]$ and $\beta \geq 0$ be the dual variables. Then, we can write down the KKT conditions as follows:

$$
\begin{cases}
\displaystyle\sum_{i=1}^{N}(a_{i2} - a_{i1})z_i + 2\beta w = 0, \\
a_{i1} + a_{i2} + a_{i3} = \dfrac{1}{N}, \forall i \leq N, \\
\kappa\displaystyle\sum_{i=1}^{N} a_{i2} + 2\beta\lambda = \epsilon, \\
a_{i1}(1 - w^T z_i - s_i) = 0, \\
a_{i2}(1 + w^T z_i - \lambda\kappa - s_i) = 0, \\
a_{i3}s_i = 0, \\
\textcolor{blue}{\beta(\|w\|_2^2 - \lambda^2) = 0.}
\end{cases}
\tag{27}
$$

Based on (27), we have

$$
\begin{aligned}
\mathbf{0} &= \sum_{i=1}^{N}(a_{i2} - a_{i1})w^T z_i + 2\beta\|w\|_2^2 \\
&= \sum_{i=1}^{N}(a_{i2} - a_{i1})w^T z_i + 2\beta\lambda^2 = \sum_{i=1}^{N}(a_{i2} - a_{i1})w^T z_i + \lambda(\epsilon - \kappa\sum_{i=1}^{N} a_{i2}) \\
&= \lambda\epsilon + \sum_{i=1}^{N} a_{i2}(w^T z_i - \lambda\kappa) - \sum_{i=1}^{N} a_{i1}w^T z_i = \lambda\epsilon + \sum_{i=1}^{N}(a_{i2} + a_{i1})(s_i - 1).
\end{aligned}
$$

Thus, we have

$$
\lambda = \frac{1}{\epsilon}\sum_{i=1}^{N}(a_{i2} + a_{i1})(1 - s_i) = \frac{1}{\epsilon}\sum_{i=1}^{N}(\frac{1}{N} - a_{i3})(1 - s_i) \leq \frac{1}{\epsilon N}\sum_{i=1}^{N}(1 - s_i) \leq \frac{1}{\epsilon}.
$$

$\square$

**Remark 6.11** *Although we focus on the case where* $q = 2$ *in this proof, we can easily extend the techniques to study the case where* $q \in \{1, \infty\}$. *We just need to modify the blue part in* (27). *Specifically, observe that* $\|w\|_1 \leq \lambda$ *is equivalent to* $Bw \leq \lambda e_{2^d}$, *where* $B$ *is the* $2^d \times d$ *matrix whose rows are all the possible arrangements of* +1*'s and* −1*'s. On the other hand,* $\|w\|_\infty \leq \lambda$ *is equivalent to* $e_i^T w \leq \lambda$, $-e_i^T w \leq \lambda$, $\forall i \in [n]$.

Subsequently, we develop a standard ADMM algorithm to address the $w$-subproblem

$$\min_{w} \frac{1}{n} \sum_{i=1}^{n} \max \left\{ 1 - w^T z_i, 1 + w^T z_i - \lambda \kappa, 0 \right\}, \text{ s.t. } \|w\|_q \leq \lambda.$$

We apply the operator splitting technique to reformulate it as

$$\min_{w,y} \quad \frac{1}{n} \sum_{i=1}^{n} \max \left\{ 1 - y_i, 1 + y_i - \lambda \kappa, 0 \right\}$$
$$\text{s.t.} \quad Zw - y = 0,$$
$$\|w\|_q \leq \lambda.$$

---

**Algorithm 3:** ADMM for solving $w$-subproblem

---

**Input:** Choose value $(w^0, y^0, g^0) \in \mathbb{R}^d \times \mathbb{R}^n \times \mathbb{R}^n$;
Initialized the penalty parameter $\rho_0$ and shrinking parameter $\gamma \geq 1$;
**Output:** $\{(w^k, y^k, g^k)\}_{k=1}^{K}$ and function value sequences;
**for** *each iteration* **do**

    /* Accelerated projected gradient algorithm, see [15, Algorithm 5]          */

    $w^{k+1} = \arg \min\limits_{\|w\|_q \leq \lambda} \left\{ \frac{\rho_k}{2} \|Zw - y^k + \frac{g^k}{\rho_k}\|_2^2 \right\}$;

    /* Closed-form update                                      */

    $y^{k+1} = \arg \min\limits_{y \in \mathbb{R}^n} \left\{ \frac{1}{n} \sum_{i=1}^{n} \max \left\{ 1 - y_i, 1 + y_i - \lambda \kappa, 0 \right\} + \frac{\rho_k}{2} \|y - Zw^{k+1} - \frac{g^k}{\rho_k}\|_2^2 \right\}$;

    /* Dual update                                              */

    $g^{k+1} = g^k + \rho_k (Zw^{k+1} - y^{k+1})$;

    $\rho_{k+1} = \gamma \rho_k$;

**end**

---

**Single-sample proximal point update for $c > 0$**    Recall that

$$\min_{w,\lambda} \quad \frac{c}{2}\|w\|_2^2 + \max \left\{ 1 - w^T z_i, 1 + w^T z_i - \lambda \kappa, 0 \right\} + \frac{1}{2\alpha}(\|w - \bar{w}\|_2^2 + (\lambda - \bar{\lambda})^2)$$
$$\text{s.t.} \quad \|w\|_q \leq \lambda.$$

Note that $\mu = \frac{\lambda}{\sqrt{1+\alpha c}}$ and $\kappa' = \kappa \sqrt{1 + \alpha c}$. The above problem can be written as

$$\min_{w,\mu} \quad \max \left\{ 1 - w^T z_i, 1 + w^T z_i - \mu \kappa', 0 \right\} + \frac{1 + ac}{2\alpha} \left( \left\| w - \frac{\bar{w}}{1 + ac} \right\|_2^2 + (\mu - \bar{\mu})^2 \right) \quad (28)$$
$$\text{s.t.} \quad \|w\|_q \leq \sqrt{1 + \alpha c}\mu.$$

Indeed, problem (28) shares the same structure as problem (2).

Table 7: Wall-clock Time Comparison on UCI Real Dataset: $\ell_1$-DRSVM, $c = 1, \kappa = 1, \epsilon = 0.1$

| | Objective Value | | | | Wall-clock time (sec) | | | |
|------|-----------|-----------|-----------|-----------|--------|--------|--------|---------|
| | M-ISG | IPPA | Hybrid | YALMIP | M-ISG | IPPA | Hybrid | YALMIP |
| a1a | 0.7871456 | 0.7871450 | **0.7871445** | 0.7871462 | **1.233** | 2.169 | 1.545 | 16.495 |
| a2a | 0.8002882 | **0.8002879** | **0.8002879** | 0.8003101 | **0.602** | 3.546 | 0.720 | 23.557 |
| a3a | 0.7826654 | **0.7826653** | **0.7826653** | **0.7826653** | 3.687 | 4.626 | **3.696** | 32.575 |
| a4a | **0.7929724** | **0.7929724** | **0.7929724** | 0.7929959 | **0.579** | 4.109 | 0.713 | 62.456 |
| a5a | 0.7852845 | 0.7852848 | **0.7852844** | 0.7853440 | **0.639** | 2.929 | 1.156 | 109.620 |
| a6a | **0.7764425** | **0.7764425** | **0.7764425** | 0.7767701 | **1.272** | 5.756 | 1.576 | 185.080 |
| a7a | 0.7822116 | **0.7822114** | 0.7822115 | 0.7827171 | **2.121** | 4.336 | 2.976 | 270.480 |
| a8a | **0.7805498** | **0.7805498** | **0.7805498** | 0.7836023 | **2.502** | 10.184 | 2.798 | 372.050 |
| a9a | 0.7767114 | **0.7767099** | 0.7767113 | 0.7791881 | **3.018** | 9.295 | 6.203 | 642.160 |

Table 8: Wall-clock Time Comparison on UCI Real Dataset: $\ell_\infty$-DRSVM, $c = 1, \kappa = 1, \epsilon = 0.1$

|  | Objective Value | | | | Wall-clock time (sec) | | | |
|  | M-ISG | IPPA | Hybrid | YALMIP | M-ISG | IPPA | Hybrid | YALMIP |
|---|---|---|---|---|---|---|---|---|
| a1a | 0.7853266 | **0.7853265** | **0.7853265** | 0.7853269 | **0.510** | 0.707 | 0.687 | 12.928 |
| a2a | 0.7987669 | 0.7987666 | 0.7987667 | **0.7987663** | **0.861** | 1.233 | 1.182 | 20.850 |
| a3a | 0.7810149 | **0.7810140** | 0.7810145 | 0.7810568 | **0.350** | 0.666 | 0.563 | 28.494 |
| a4a | **0.7913534** | **0.7913534** | **0.7913534** | 0.7913963 | **0.540** | 1.100 | 0.679 | 63.037 |
| a5a | 0.7836246 | **0.7836189** | 0.7836246 | 0.7836478 | **0.737** | 1.421 | 0.897 | 96.314 |
| a6a | 0.7748542 | **0.7748533** | 0.7748537 | 0.7759606 | **1.203** | 2.250 | 1.763 | 201.510 |
| a7a | **0.7806894** | **0.7806894** | **0.7806894** | 0.7811091 | **1.753** | 3.244 | 2.124 | 370.510 |
| a8a | 0.7789801 | **0.7789800** | **0.7789800** | 0.7850996 | **2.390** | 4.568 | 2.981 | 365.140 |
| a9a | **0.7750633** | **0.7750633** | **0.7750633** | 0.7776798 | **3.368** | 6.706 | 4.371 | 753.330 |

**Remark 6.12** *For $\ell_2$-DRSVM problems, it is worth mentioning that the $c > 0$ case is identical to the $c = 0$ case from a modeling perspective, which depends on the different robustness levels $\epsilon$ and $\kappa$.*