[Reviews · NeurIPS 2020]

Review 1

Summary and Contributions: This paper studies the Wasserstein distributionally robust support vector machine problems and proposes two efficient methods to solve them. Convergence rates are established by the Holderian growth condition. The updates in each iteration of these algorithms can be computed efficiently, which is the focus of this paper.

Strengths: I did not know the Wasserstein distributionally robust optimization before. So I am not sure whether the problem studied in this paper is significant. 1. Two algorithms are proposed for the DRO problem, one is based on the projected subgradient algorithm and the other is based on the proximal point algorithm. The update in each iteration is implemented efficiently 2. Convergence rates are given by establishing the Holderian growth condition.

Weaknesses: 1. In Table 1, the convergence when c=0 is faster than the one when c>0. However, when c>0, the objective has an additional quadratic term than the case of c=0. From my experimence, adding a quadratic term always makes the algorithms faster. So Table 1 looks strange to me. It might better to give more intuitions and explanations. The sharpness condition may give a faster convergence than the QG condition. However, can the authors prove the convergence beyond the QG condition when c>0, for example, give a possible faster rate by exploiting the specification of the problem? 2. In Section 5.1, the authors claim that IPPA is slower than M-ISG. However, from my experience, the proximal point algorithm is always faster than the subgradient method in general. It might better to explain more. 3. Please confirm the Holderian growth condition is global or local.

Correctness: Correct

Clarity: Yes

Relation to Prior Work: Yes

Reproducibility: Yes

Additional Feedback: After rebuttal: Thanks for the response. It has addressed my questions.


Review 2

Summary and Contributions: The authors propose first order algorithms to solve Wasserstein distributionally robust support vector machine problems. The proposed approach heavily exploits the hidden structure of the resulting reformulation. In particular, when the transportation cost is the Euclidean norm, the authors show that they can solve the sub-problems in projected subgradient and proximal point algorithms analytically. Moreover, when the transportation cost is the \ell_1 or \ell_infty norm, they reformulate the sub-problems as a one dimensional optimization problems, which can be efficiently solved with a modified version of the secant algorithm. ================================= [Update after reading rebuttal]: My evaluation remains the same.

Strengths: I found the technical analysis of the paper very interesting. Numerical results also show that the proposed algorithms consistently outperform general convex optimization solvers by a great margin.

Weaknesses: The only weakness of the paper, in my opinion, is that the algorithms are mainly designed to handle the support vector machine problem. The extension of the proposed algorithm to more general piece-wise linear cost functions is very interesting, but I can understand that the authors have to restrict the problem setting due to lack of space or difficulty in subsequent analyses.

Correctness: I have checked appendix in the paper, and to the best of my knowledge, all proofs are correct. The empirical results are also consistent with theory.

Clarity: The paper is very well-written and well-organized. All steps are clearly explained and the motivation is crystal clear.

Relation to Prior Work: The focus of the current work is support vector machine, and its connection to the previous results are clearly made in the paper.

Reproducibility: Yes

Additional Feedback:


Review 3

Summary and Contributions: The authors have proposed two new algorithms to solve the timely and important distributionally robust SVM problem. These new algorithms are instances of incremental projected subgradient descent and incremental proximal point algorithm. The main novelty of ISG and IPPA proposed in this work is that they developed efficient algorithms (in linear time) for the subproblems in these solving strategies, which are interesting and potentially beneficial for other problems with similar structures. Besides, the authors carefully analyze the iteration complexity of their new algorithms under the so-called BLR condition. They show if the BLR condition is satisfied, the exponent in the Holderian growth condition could be explicitly determined. Thus, the proposed algorithms enjoy sublinear or linear fast rates in these settings. The empirical speedups appear to be substantial. Overall, this paper contributes a well-founded and empirically successful algorithms for accelerating solution of the distributionally robust SVM problem. This could be impactful for both practitioners and researchers, as it extends DRO methodology to large datasets in real machine learning tasks.

Strengths: 1. Fast solution of DRSVM problem is potentially useful and (to my knowledge) previously unaddressed. This work proposed two novel incremental algorithms (incremental projected subgradient descent and incremental proximal point algorithm) to tackle the DRSVM problem. 2. The efficient solution proposed to address the single proximal point update is interesting and highly non-trivial. To the best of my knowledge, this class of problem has not been well addressed before. 3. The convergence rates of the new algorithms are analyzed under the BLR condition for the set of optimal solutions. The theorem shows the new algorithms enjoy sublinear/linear rates, which is unknown before. 4. They conducted extensive experiments that show the newly proposed strategies significantly outperform existing approaches. 5. The authors do a quite good job of the literature review and provide detailed comparisons between theirs and existing techniques. The paper is fairly easy to follow, with the exception of the citation mismatch between main paper and appendix.

Weaknesses: 1. I think the authors might want to provide more details on the Bound Linear Regularity condition in Def.2 as BLR is the central assumption in the main Thm 4.3. I am especially interested in how strong is this assumption? That is to say, can we have some simple examples that satisfy BLR? For example, if g is strongly convex, is arg_x g(Ax) BLR? 2. In Figure 4, dataset a3a, it seems that the Hybrid strategy does not run as fast as GS-ADMM and YALMIP. Is there some reason for that? I am interested in that as in all the other setting the Hybrid run really fast. 3. It might be better if the authors could make their fast solvers available to the practitioners and researchers in the community. Minor points: 1. Figure 1: larger font and legend makes the figure clear 2. In Appendix, there might be some mismatching of the number of references, e.g., [15]->[16], [25]->[26].

Correctness: Seems correct to me.

Clarity: The paper is well written and in good style. Some typos should be fixed.

Relation to Prior Work: Good. The authors do a quite good job of the literature review and provide detailed comparisons between theirs and existing techniques.

Reproducibility: Yes

Additional Feedback: == After Response == Thank you to the authors for the response and I am satisfied with that.


Review 4

Summary and Contributions: This paper presents two efficient optimization algorithms for Wasserstein distributionally robust support vector machine. The authors propose to jointly optimize the learning parameter w and its l_q-norm upper bound \lambda, where l_1, l_2, and l_infty norm-induced transport costs of w are considered. The two optimization methods are based on epigraphical projection-based incremental algorithms: one is incremental projected subgradient descent (ISG) method, and the other is incremental proximal point algorithm (IPPA). Theoretical analysis is shown for ISG and IPPA on linear/sublinear convergence rates. ================ [After reading the rebuttal] I have read the authors' feedback and other reviews. I agree with other reviewers on the theoretical contributions of this paper, and the rebuttal addressed my previous concerns. I updated my score from 5 to 6 accordingly.

Strengths: The proposed method is well motivated and theoretically grounded. The authors provide both theoretical proof and empirical results to support the convergence behavior of both ISG and IPPA. The paper is well written. The summary on the properties of DRSVM problem under different conditions is clear, and the proposed incremental algorithms for improving optimization efficiency is valid.

Weaknesses: My major concern is on the empirical results. Maybe I missed it, but it seems not quite clear to me why the comparing methods are selected under different conditions. For example, why GS-ADMM results are not shown for l_1 and l_2-norm optimization, i.e., Table 2 and 3? It would be helpful to show the objective function curves of the comparing method in Fig. 1. How the parameters c, κ, and \epsilon determined in the experiments?

Correctness: The claims and method look correct to me. More implementation details can be provided to make the empirical results more convincing.

Clarity: The paper is well written.

Relation to Prior Work: The difference from previous contributions is discussed.

Reproducibility: No

Additional Feedback:

[Author Response · NeurIPS 2020]

We thank the useful suggestions from the reviewers. Below please find our responses to the major points raised.

**To Review 1: About the background:** W-DRO problems have received much attention in the machine learning

community as they not only provide a probabilistic justification of existing regularization techniques but also offer

a powerful alternative approach to tackle ERM problems; see [22]. However, the only known way to solve most

existing DRO formulations is to use general-purpose solvers, which limits the scalability of the approach. Our work is

motivated by the desire to develop practically efficient methods with provable guarantees for DRO problems, so as to

realize the benefits of the approach in large-scale learning settings.

**Q1:** Table 1 looks strange to me. It might better to give more intuitions and explanations.

**R1:** It may be not true that adding a quadratic term always makes the algorithms faster. In a nutshell, the concrete

convergence rate depends on the algorithm used and whether it can exploit the regularity condition of the problem at

hand. For instance, $f(x) = |x|$ satisfies the sharpness condition, and the subgradient method (SubG) can achieve a

linear rate. However, if we add a quadratic term $g(x) = |x| + x^2$, the sharpness condition no longer holds but QG

holds. In this case, SubG is only known to converge at the sublinear $\mathcal{O}(\frac{1}{k})$ rate but proximal gradient descent (PGD)

is known to converge linearly. However, PGD cannot be applied to our setting mainly due to the non-separability of

the non-smooth objective. In particular, the associated proximal mapping cannot be efficiently implemented.

**Q2:** Can the authors prove the convergence beyond the QG condition when $c > 0$, give a possible faster rate?

**R2:** The reviewer raises an interesting question. QG is a rather general regularity condition, which has equivalent

relationships with the error bound and PL/KL properties. To the best of our knowledge, almost all problem-specific

convergence analyses utilize either a QG- or PL/KL-type regularity condition, including ours (both sharpness and QG

can be viewed as a KL-type condition). It is not clear whether our problem possesses some more general regularity

conditions, especially for the $\lambda$ variable. Thanks for your question.

**Q3**: From my experience, the proximal point algorithm is always faster than the SubG in general.

**R3**: If the cost of proximal point update is comparable with SubG, PPA is indeed faster than the SubG (i.e., $n = 1$ in

our paper). However, this is not the case for incremental methods. As we stated in lines 258-268, the main reason is

that IPPA can only update one sample at a time. Thus, if we take batch size = 1 for M-ISG, we can see that IPPA still

enjoys substantial advantages over ISG, see Fig.1 (a)-(d). Nevertheless, M-ISG can update the batch data at once and

thus less epigraphical projection operations are required for each epoch. Moreover, nested for-loop is not so efficient

in MATLAB. Thus, IPPA is slower than M-ISG w.r.t the Wall-clock Time. Thanks for your comments.

**Q4**: The Holderian growth condition is global or local? **R4**: **Global**! Thanks for your kind reminder to clarify this.

**To Review 2:** Thank you for pointing out the interesting research direction. We will explore this in future work.

**To Review 3: Q1**: More details on the BLR condition...? **R1**: BLR is a classic assumption in variational analysis, see

section 3.3 in [28]. In your example $\min g(Ax)$, the optimal set can be characterized by a linear system $\{x : Ax = y^*\}$

for some fixed $y^*$, which is polyhedral and satisfies the BLR automatically. More examples can be found in [2,28].

**Q2**: For dataset a3a, it seems that the Hybrid strategy does not run as fast as GS-ADMM and YALMIP.

**R2:** Thanks for your question. The main reason is that the convergence rate of incremental methods depends on the

sharpness constant, which in turn is data-dependent (i.e., condition number). You can also observe that a8a is slower

that a9a, but a9a has a large problem size. We have to emphasize that both GS-ADMM and YALMIP do not scale well

with problem size. Thus, you can observe that the performance gap grows considerably as the problem size increases.

**To Review 4:** Thanks for your comments. We have to emphasize that you can certainly reproduce all experiment

results based on the details provided in the Appendix (i.e., Table 5,6; Algorithm 1,2,3). We will release our code later.

**Q1:** ... For example, why GS-ADMM results are not shown for $l_1$ and $l_2$-norm optimization?

**R1:** Thanks for your kind reminder. As we stated in lines 247-249, we just extend GS-ADMM to tackle the $\ell_\infty$

DRSVM problem. Our major concern is that [16] only provides the source code to deal with the $\ell_\infty$ case. For the

sake of fairness, we only report this situation. Moreover, the $\ell_\infty$ case is the most efficient (i.e., the faster inner solver -

conjugate gradient with an active set method can only tackle the $\ell_\infty$ case in [16]). Thus, Table 4 is enough to verify the

efficiency of our proposed method. Based on your advice, we will add more comparison experiments in the revision.

**Q2:** It would be helpful to show the objective function curves of the comparing method in Fig. 1.

**R2:** Thanks for your suggestion. Let us explain a little more to clarify. First, Fig.1 aims to corroborate the theoretical

finding in Table 1. Namely, we want to showcase the empirical performance of ISG and IPPA under different regularity

conditions and step size strategies. This is why we only report the curve of incremental methods. Second, the YALMIP

solver relies on interior-point algorithms, which use second-order (Hessian) information. It is thus not entirely fair to

compare the function value curves of our first-order method and the YALMIP solver. For GS-ADMM, it requires an

outer loop to search for the $\lambda^*$ (i.e., like a two-stage algorithm). Hence, it is also incomparable to ours in terms of

function value. Third, since the problem is convex, all methods will converge to the same objective value.

**Q3:** How the parameters $c$, $\kappa$, and $\epsilon$ determined in the experiments?

**R3:** We mentioned the hyperparameter setting in our original paper (i.e., line 255 and all table titles). $\kappa$ and $\epsilon$ do not

affect the convergence rate/computational complexity, see Theorem 4.3 for details. The constant $c$ only influence the

regularity condition. Also, some preliminary experiments have been conducted for different hyperparameter sets and

all experiment performances have no obvious difference. Thus, we just follow the experimental setup in [16].

[Meta-Review · NeurIPS 2020]

This paper addresses important problems, presents novel techinical analysis, and neumerical results are encouraging. All of the reviewers and I have agreed to accept this paper. However, several reviewers (especially R1 and R4) have pointed out some important concerns. Please consider revising your paper to address them before submitting a camera-ready.